# Validity and effectiveness of paediatric early warning systems and track and trigger tools for identifying and reducing clinical deterioration in hospitalised children: a systematic review

Rob Trubey,[1] Chao Huang,[2] Fiona V Lugg-Widger,[1] Kerenza Hood,[1] Davina Allen,[3] Dawn Edwards,[4] David Lacy,[5] Amy Lloyd,[1] Mala Mann,[6] Brendan Mason,[7] Alison Oliver,[8] Damian Roland,[9,10] Gerri Sefton,[11] Richard Skone,[8] Emma Thomas-Jones,[1] Lyvonne N Tume,[12] Colin Powell[13,14]

**Correspondence to**
Dr Rob Trubey;
trubeyrj@cf.ac.uk

## ABSTRACT

**Objective** To assess (1) how well validated existing paediatric track and trigger tools (PTTT) are for predicting adverse outcomes in hospitalised children, and (2) how effective broader paediatric early warning systems are at reducing adverse outcomes in hospitalised children.

**Design** Systematic review.

**Data sources** British Nursing Index, Cumulative Index of Nursing and Allied Health Literature, Cochrane Central Register of Controlled Trials, Database of Abstracts of Reviews of Effectiveness, EMBASE, Health Management Information Centre, Medline, Medline in Process, Scopus and Web of Knowledge searched through May 2018.

**Eligibility criteria** We included (1) papers reporting on the development or validation of a PTTT or (2) the implementation of a broader early warning system in paediatric units (age 0–18 years), where adverse outcome metrics were reported. Several study designs were considered.

**Data extraction and synthesis** Data extraction was conducted by two independent reviewers using template forms. Studies were quality assessed using a modified Downs and Black rating scale.

**Results** 36 validation studies and 30 effectiveness studies were included, with 27 unique PTTT identified. Validation studies were largely retrospective case-control studies or chart reviews, while effectiveness studies were predominantly uncontrolled before-after studies. Metrics of adverse outcomes varied considerably. Some PTTT demonstrated good diagnostic accuracy in retrospective case-control studies (primarily for predicting paediatric intensive care unit transfers), but positive predictive value was consistently low, suggesting potential for alarm fatigue. A small number of effectiveness studies reported significant decreases in mortality, arrests or code calls, but were limited by methodological concerns. Overall, there was limited evidence of paediatric early warning system interventions leading to reductions in deterioration.

**Conclusion** There are several fundamental methodological limitations in the PTTT literature, and the predominance of single-site studies carried out in specialist centres greatly limits generalisability. With limited evidence of effectiveness, calls to make PTTT mandatory across all paediatric units are not supported by the evidence base.

**PROSPERO registration number** CRD42015015326

## Strengths and limitations of this study

► Paediatric early warning systems and paediatric track and trigger tools (PTTT) are increasingly used by paediatric units across Europe, North America, Australia and elsewhere—this study is a timely review of the evidence for their validity and effectiveness.

► A comprehensive search was carried out across multiple databases and included published as well as grey literature.

► The review highlights methodological weaknesses and gaps in the current evidence base and makes suggestions for future research.

► Heterogeneity in study populations, study designs and outcome measures make it difficult to compare and synthesise findings across the wide range of early warning systems and PTTT being used in practice.

► The review is limited in scope to quantitative validation and effectiveness studies, so must be considered alongside wider literature reflecting on potential secondary benefits of early warning systems and PTTT for communication, teamwork and empowerment.

## BACKGROUND

Failure to recognise and respond to clinical deterioration in hospitalised children is a major safety concern in healthcare. The underlying causes of this problem are

clearly multifactorial,[1–3] but paediatric 'early warning systems' have been strongly advocated as one approach to improving recognition of deterioration in paediatric units.[1 2 4]

A paediatric 'early warning system' can be considered any patient safety initiative or programme which aims to monitor, detect and respond to signs of deterioration in hospitalised children in order to avert adverse outcomes and premature death. Such systems are often multifaceted and may include the use of rapid response teams (RRT) or medical emergency teams (MET), education or training to improve clinical staff's ability to identify deterioration or strategies aimed at improving staff communication and situational awareness.

An increasingly commonplace paediatric 'early warning system' initiative is the use of a 'track and trigger tool': these tools, also commonly used in adult care, provide a formal framework for evaluating routine physiological, clinical and observational data for early indicators of patient deterioration. They are typically integrated into routine observation charts or electronic health records and compare patient observations with predefined 'normal' thresholds. When one or more observation is considered abnormal, staff are directed to various clinical actions, including but not limited to altered frequency of observations, review by senior staff or more appropriate treatment or management. Tools may be paper based or electronic and monitoring may be automated or manually undertaken by staff.

These tools have been referred to in the literature using a number of different terms: paediatric early warning scores (PEWS); paediatric early warning tools (PEWT), track and trigger tools (TTT) and many others. Here, we refer to the tools themselves using the term 'paediatric track and trigger tools' (PTTT). A variety of PTTT have been developed, typically by teams based in specialist paediatric centres and often used as a means of triggering a dedicated response team. Their advocacy has recently led to widespread uptake across a variety of different paediatric units, including many non-specialist centres where patient populations and resources may differ. In the UK, a recent cross-sectional survey found that 85% of paediatric units were using some form of PTTT, most of which were non-specialist centres without a dedicated response team.[5] Despite their widespread use, recent reviews have questioned the evidence base for their effectiveness in improving patient outcomes.[6 7] The current review aimed to build on this work, assessing in depth the evidence base for both the validity of PTTT for predicting in-patient deterioration and the effectiveness of broader 'early warning systems' at reducing instances of mortality and morbidity in paediatric settings:

► Question 1: how well validated are existing PTTT and their component parts for predicting inpatient deterioration?
► Question 2: how effective are paediatric early warning systems (with or without a PTTT) at reducing mortality and critical events?

## METHODS

This systematic review is reported in accordance with the Preferred Reporting Items for Systematic Review and Meta-Analyses (PRISMA) guidelines.[8] Our review protocol is registered with the PROSPERO database CRD42015015326.

### Search strategy

A comprehensive search was conducted across a range of databases to identify relevant studies in the English language. Published and unpublished literature was considered where publicly available, as were studies in press. The following databases were searched through May 2018: British Nursing Index, Cumulative Index of Nursing and Allied Health Literature, Cochrane Central Register of Controlled Trials, Database of Abstracts of Reviews of Effectiveness, EMBASE, Health Management Information Centre, Medline, Medline in Process, Scopus and Web of Knowledge (Science Citation Indexes). To identify additional papers, published, unpublished or research reported in the grey literature, a range of relevant websites and trial registers were searched including Clinical Trials.gov. To identify published papers that had not yet been catalogued in the electronic databases, recent editions of key journals were hand-searched. The search terms included 'early warning scores', 'alert criteria', 'rapid response', 'track and trigger' and 'early medical intervention' (see online supplementary table 1).

### Eligibility screening and study selection

PICOS parameters guided inclusion criteria for the validation and effectiveness studies (see online supplementary table 2). Papers reporting development of validation of a PTTT were included for question 1, whereas papers reporting the implementation of any broader 'paediatric early warning system' (with or without a PTTT) were eligible for question 2. Both research questions were limited to studies that involved inpatients aged 0–18 years. Outcome measures considered were mortality and critical events, including: unplanned admission to a higher level of care, cardiac arrest, respiratory arrest, medical emergencies requiring immediate assistance, children reviewed by paediatric intensive care unit (PICU) staff on the ward (in specialist centres) or reviewed by external PICU staff (for non-specialist centres), acuity at PICU admission and PICU outcomes. A range of study designs were considered for both questions.

Two of the review authors independently screened the titles and abstracts yielded in the search. Full texts were reviewed independently by six reviewers against the above eligibility criteria and were assigned to the relevant review question if included. Reasons for exclusion were recorded. Separate data extraction forms were developed for validation and effectiveness studies. The forms had common elements (study design, country, setting, study population, description of the PTTT or early warning system, statistical techniques used, outcomes assessed). Additional data items for validation studies included

the items in the PTTT, modifications to the PTTT from previous versions, predictive ability of individual items and the overall tool, sensitivity and specificity and inter-rater and intra-rater reliability. Effectiveness studies included an assessment of outcomes in terms of mortality and various morbidity variables. Data extraction was carried out by two reviewers and discrepancies were resolved by discussion. For effectiveness studies, effect sizes and 95% CIs were calculated or reported as risk ratios (RR) or ORs as appropriate, with p values reported to assess statistical significance. Data analysis was conducted using an online medical statistics tool.

## Quality appraisal

Methodological quality and risk of bias was assessed for each included study using a modified version of the Downs and Black rating scale[9] (templates shown in online supplementary table 3).

## Patient and public involvement

This review was conducted as part of a larger mixed-methods study (ISRCTN94228292), which used a formal, facilitated parental advisory group. The group comprised parents of children who had experienced an unexpected adverse event in a paediatric unit and provided input which helped to shape the broader research questions and outcome measures. The results of the review will be disseminated to parents through this group.

## RESULTS

Figure 1 shows the PRISMA flow diagram for both research questions.

## Study characteristics

Table 1 summarises the study characteristics of validation and effectiveness papers in the review.

## Types of PTTS and components

Across 66 studies, we identified 27 unique PTTT (table 2). Twenty PTTTs were based on one of four different tools: Monaghan's Brighton PEWS,[10] the Bedside PEWS,[11] the Bristol PEWT[12] and the Melbourne Activation Criteria (MAC).[13] Other PTTT described in the literature included the National Health Service Institute for Innovation and Improvement (NHS III) PEWS[14] (the second most commonly used PTTT in UK paediatric settings[5]), RRT and MET activation criteria[15–18] and one prediction algorithm developed from a large dataset of electronic health data.[19]

Table 2 illustrates the range of physiological and behavioural parameters underpinning PTTT. Common parameters included heart rate (present in 26 out of 27 PTTT), respiratory rate (24), respiratory effort (24) and level of consciousness or behavioural state (24). All PTTT required at least six different parameters to be collected.

## Question 1: how well validated are PTTT and component parts for predicting inpatient deterioration?

Nine validation papers meeting inclusion criteria were excluded from analysis: eight did not report any

performance characteristics of the PTTT for predicting deterioration[20–27] and one study calculated incorrect sensitivity/specificity outcomes[12] (see online supplementary table 4). The remaining 27 validation studies, evaluating the performance of 18 unique PTTT, are described in table 3. Four studies evaluated multiple PTTTs[3 19 28 29] and one paper described three separate studies of the same PTTT.[30]

Five cohort studies were included,[14 31–34] three based on the same dataset. All other studies were either case-control or chart reviews. Thirteen papers implemented the PTTT in practice,[23 30 31 34–43] while the remaining studies 'bench tested' the PTTT—researchers retrospectively calculated the score based on data abstracted from medical charts and records. All studies were conducted in specialist centres with only one multicentre study reported.[44]

## Outcome measures

PTTT were evaluated for their ability to predict a wide range of clinical outcomes. Composite measures were used in 8 studies,[14 23 29 32 33 37 45 46] cardiac/respiratory arrest or a 'code call' was used (singularly or part of a composite outcome) in 6 studies,[23 28 29 37 45 47] while 22 studies used transfer to a to PICU or paediatric high-dependency unit as the main outcome.[3 11 19 23 28–34 36 37 39 41–44 46 48 49]

## Predictive ability of individual PTTT components

Three validation papers reported on the performance characteristics of individual components of the tool for predicting adverse outcomes.[11 33 42] Parshuram *et al*, for instance, reported area under the receiver operating characteristic curve (AUROC) values for individual PTTT items of a pilot version of the Bedside PEWS: ranging from 0.54 (bolus fluid) to 0.81 (heart rate), compared with 0.91 for the overall PTTT.[11] All other studies reported outcomes for the PTTT as a whole.

## PEWS score

The predictive ability of the 16-item PEWS score was assessed by one internal[47] (AUROC=0.90) and two external case-control studies[28 29] (AUROC range=0.82–0.88) with a range of outcome measures and scoring thresholds. One case-control study used an observed prevalence rate to calculate a positive predictive value (PPV) of 4.2% for the tool in predicting code calls[47] (for every 1000 patients triggering the PTTT, 42 would be expected to deteriorate).

## Bedside PEWS and derivatives

The Bedside PEWS was evaluated in one internal[11] (AUROC=0.91) and five external case-control studies[19 28 29 44 46] (AUROC range=0.73–0.90) for a range of different outcome measures and at different scoring thresholds. One case-control study calculated a PPV of 2.1% for identifying children requiring urgent PICU transfer within 24 hours of admission, based on locally observed prevalence rates.[19] A modified version of the Bedside PEWS (with temperature added) demonstrated an AUROC of 0.86 in an external case-control study with

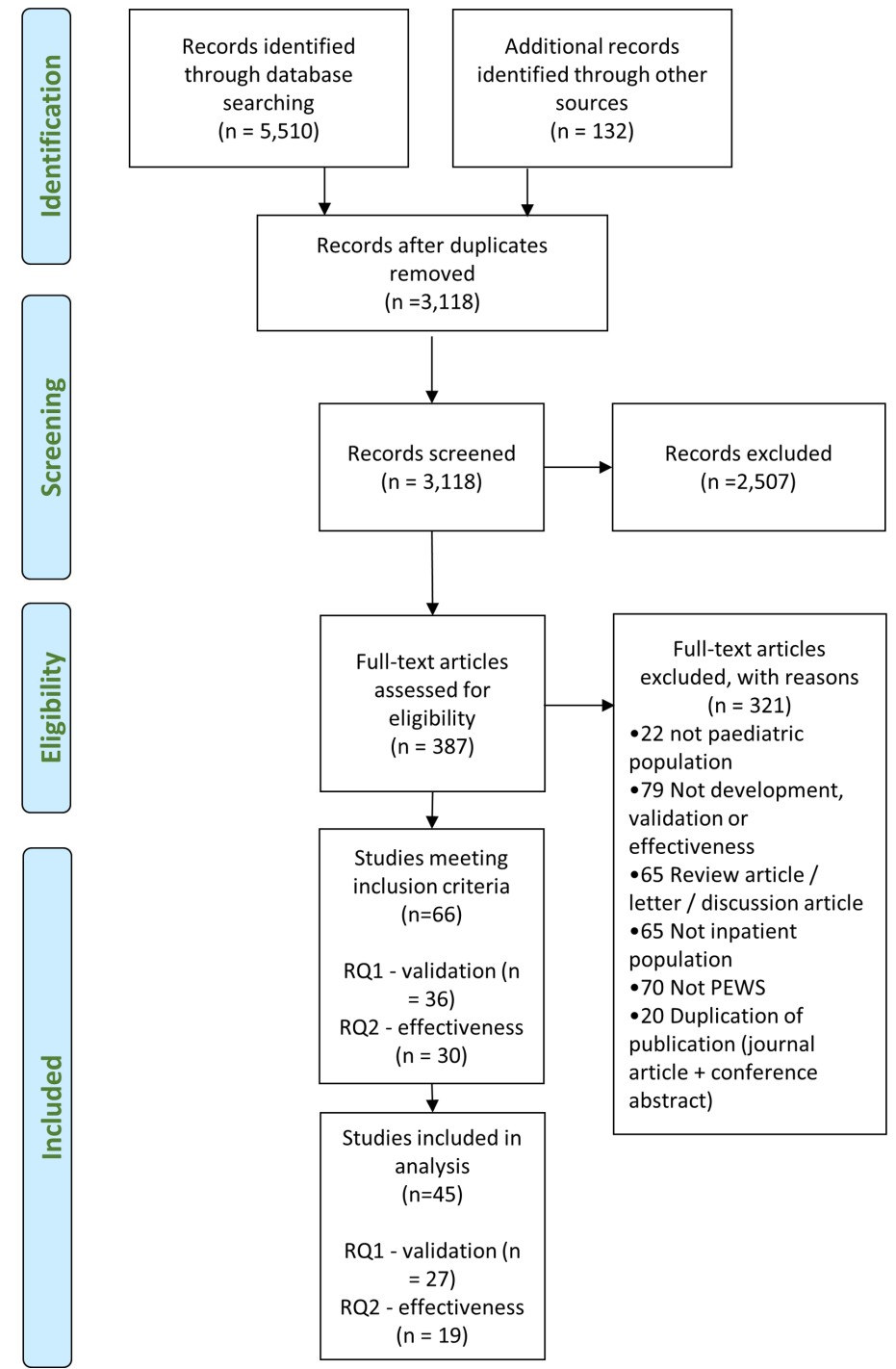

**Figure 1** Preferred Reporting Items for Systematic Review and Meta-Analyses flow diagram of study inclusion. PEWS, paediatric early warning scores.

a composite outcome of death, arrest or unplanned PICU transfer.[29]

### Brighton PEWS and derivatives

Six different PTTT based on the original Brighton PEWS were evaluated across 11 studies.[19 29 31 37 39–42 45 48 50] The Modified Brighton PEWS (a) was evaluated for its ability to predict PICU transfers in one large prospective cohort study (AUROC=0.92, PPV=5.8%),[31] and an external case-control study tested the same score for predicting

urgent PICU transfers within 24 hours of admission (AUROC=0.74, PPV=2.1%).[19]

An external case-control study used a composite measure of death, arrest or PICU transfer to evaluate the Modified Brighton PEWS (b) (AUROC=0.79) and the Modified Brighton PEWS (d) (AUROC=0.74).[29] The latter tool was evaluated in a further internal case-control study for predicting PICU transfer (AUROC=0.82).[48]

**Table 1** Summary of the study characteristics of the 36 validation (question 1) and 30 effectiveness (question 2) papers included in the review

| Validation studies (n=36) | N (%) | Effectiveness studies (n=30) | n (%) |
|---|---|---|---|
| Type | | Type | |
| Full text | 22 (61.1) | Full text | 21 (70.0) |
| Abstract | 14 (38.9) | Abstract | 9 (30.0) |
| Country | | Country | |
| USA | 15 (41.7) | USA | 18 (60.0) |
| UK | 12 (33.3) | UK | 3 (10.0) |
| Canada | 2 (5.5) | Canada | 2 (6.7) |
| Australia | 0 (0.0) | Australia | 3 (10.0) |
| Other | 5 (13.9) | Other | 3 (10.0) |
| Multiple | 1 (2.8) | Multiple | 1 (3.3) |
| Unclear | 1 (2.8) | Unclear | 0 (0.0) |
| Year of study | | Year of study | |
| Pre-2012 | 10 (27.8) | Pre-2012 | 15 (50.0) |
| 2012 | 3 (8.3) | 2012 | 1 (3.3) |
| 2013 | 6 (16.7) | 2013 | 2 (6.7) |
| 2014 | 5 (13.9) | 2014 | 6 (20.0) |
| 2015 | 7 (19.4) | 2015 | 0 (0.0) |
| 2016 | 2 (5.6) | 2016 | 2 (6.7) |
| 2017 | 3 (8.3) | 2017 | 1 (3.3) |
| 2018 | 0 (0.0) | 2018 | 3 (10.0) |
| Setting | | Setting | |
| Specialist/tertiary | 33 (91.7) | Specialist/tertiary | 29 (96.7) |
| Non-specialist/community | 0 (0.0) | Non-specialist/community | 1 (3.3) |
| Unclear | 3 (8.3) | Unclear | 0 (0.0) |
| Single-centre/multicentre | | Single-centre/multicentre | |
| Single-centre | 35 (97.2) | Single-centre | 28 (93.3) |
| Multicentre | 1 (2.8) | Multicentre | 2 (6.7) |
| Study population | | Study population | |
| General inpatients | 23 (63.9) | General inpatients | 20 (66.6) |
| Specialist population | 11 (30.6) | Specialist population | 5 (16.7) |
| Unclear | 2 (5.6) | Unclear | 5 (16.7) |
| Study design | | Study design | |
| Case-control | 18 (50.0) | Uncontrolled before-after | 26 (86.7) |
| Case/chart review | 10 (27.8) | Controlled before-after | 1 (3.3) |
| Cohort | 7 (19.4) | Interrupted time series | 2 (6.7) |
| Pilot study | 1 (2.8) | Cluster randomised trial | 1 (3.3) |

The Children's Hospital Early Warning Score (CHEWS) had a reported AUROC of 0.90 for predicting PICU transfers or arrests in a large internal case-control study.[50] A modification for cardiac patients, the Cardiac CHEWS (C-CHEWS) was evaluated by one internal study on a cardiac unit[37] (AUROC=0.90) looking at arrests or unplanned PICU transfers, and two external studies of oncology/haematology units[41 42] for the same outcome (AUROC=0.95). Finally, the Children's Hospital

Los Angeles PEWS was evaluated by in a small internal case-control study for prediction of re-admission to PICU after initial PICU discharge[40] (AUROC=0.71).

### MAC and derivatives

The MAC was assessed by one external case-control study with an outcome of death, arrest or unplanned PICU transfer[29] (AUROC=0.71) and a large external cohort study with an outcome of death or unplanned

## Table 2  Summary of PTTTs

| PTTT name (references) | Development/ modification details | Score/ trigger | Choice of thresholds/ parameters | Age-dependent thresholds | No. of items in the tool* | PTTT parameters | | | | | | | | | | | | | | | Other items |
|---|---|---|---|---|---|---|---|---|---|---|---|---|---|---|---|---|---|---|---|---|---|
| | | | | | | Respiratory rate | Heart rate | Respiratory effort/ distress | LOC/ behaviour | Oxygen saturation | Capillary refill time | Oxygen therapy | Systolic blood pressure | Pain | Staff concern | Skin colour | Airway problems | Temperature | Pulses | Family concern | |
| **Paediatric early warning system (PEWS) score and derivatives** | | | | | | | | | | | | | | | | | | | | | |
| PEWS score[28 47] | Developed for use in Canadian tertiary centre.[47] Nurse-generated candidate items reduced by focus groups/Delphi and evaluation with clinical dataset (code blue calls, n=87; controls, n=128). Development and validation datasets not independent. | Score | Expert opinion | Yes | 16 | ✓ | ✓ | ✓ | ✓ | ✓ | ✓ | ✓ | ✓ | | | | ✓ | ✓ | ✓ | | Bolus fluid, medications, home oxygen, any previous admission to an ICU, central venous line in situ, transplant recipient, severe cerebral palsy, gastrostomy tube, greater than three medical specialties involved in care. |
| Bedside PEWS[11 19 25 26 28 44 46 59 64 65 70] | Developed for use in US tertiary centre.[11] Routinely collected items assessed for discriminatory ability using clinical dataset (PICU admission, n=60; controls, n=120). Development and validation set not independent. | Score | Expert opinion | Yes | 7 | ✓ | ✓ | ✓ | | ✓ | ✓ | ✓ | ✓ | | | | | | | | |
| Modified Bedside PEWS (a)[60] | Modification to Bedside PEWS for use in Dutch tertiary centre. Added temperature; modified wording of respiratory effort and oxygen therapy items. | Score | Expert opinion | Yes | 8 | ✓ | ✓ | ✓ | | ✓ | ✓ | ✓ | ✓ | | | | | ✓ | | | |
| Modified Bedside PEWS (b)[49] | Modification to Bedside PEWS for use in US tertiary centre. Changed normal thresholds for HR and RR based on analysis of local clinical data. | Score | HR/RR data driven | Yes | 7 | ✓ | ✓ | ✓ | | ✓ | ✓ | ✓ | ✓ | | | | | | | | |
| **Brighton PEWS and derivatives** | | | | | | | | | | | | | | | | | | | | | |
| Brighton PEWS[10 54] | Initial development for use in UK tertiary centre. Adapted from existing adult scores, but amended based on local clinical consensus. Small audit of patients (n=30) described but no formal validation. | Score | Expert opinion | No | 5 | ✓ | ✓ | ✓ | ✓ | | ✓ | ✓ | | | | ✓ | | | | | Quarter hourly nebulisers, persistent vomiting postsurgery. |
| Modified Brighton PEWS (a)[19 31 39] | Modification of Brighton PEWS for use in general medical ward of a US tertiary centre. Altered thresholds for oxygen therapy; changed wording for respiratory effort; modified escalation algorithm. | Score | Expert opinion | No | 5 | ✓ | ✓ | ✓ | ✓ | | ✓ | ✓ | | | | ✓ | | | | | Quarter hourly nebulisers, persistent vomiting postsurgery. |
| Modified Brighton PEWS (b)[45 72] | Modification of Brighton PEWS for use in US tertiary centre. Added age-dependent thresholds for HR and RR. | Score | Expert opinion | Yes | 5 | ✓ | ✓ | ✓ | ✓ | | ✓ | ✓ | | | | ✓ | | | | | Quarter hourly nebulisers, persistent vomiting postsurgery. |

Continued

**Table 2** Continued

| PTTT name (references) | Development/ modification details | Score/ trigger | Choice of thresholds/ parameters | Age-dependent thresholds | No. of items in the tool* | PTTT parameters | | | | | | | | | | | | | | | Other items |
|---|---|---|---|---|---|---|---|---|---|---|---|---|---|---|---|---|---|---|---|---|---|
| | | | | | | Respiratory rate | Heart rate | Respiratory effort/ distress | LOC/ behaviour | Oxygen saturation | Capillary refill time | Oxygen therapy | Systolic blood pressure | Pain | Staff concern | Skin colour | Airway problems | Temperature | Pulses | Family concern | |
| Modified Brighton PEWS (c)[52] | Modification of Brighton PEWS for use in a US haematology/oncology unit. Altered thresholds; changed respiratory effort wording; modified escalation algorithm; added and removed items. No formal validation study reported. | Score | Expert opinion | No | 3 | ✓ | ✓ | ✓ | ✓ | | ✓ | ✓ | | | | ✓ | | | | ✓ | |
| Modified Brighton PEWS (d)[48] | Modification of Brighton PEWS for use in a US tertiary centre. Modified wording of behaviour component, added age-dependent thresholds for HR and RR; removed nebulisers and persistent vomiting. | Score | Expert opinion | Yes | 3 | ✓ | ✓ | ✓ | ✓ | ✓ | ✓ | ✓ | | | | ✓ | | | | | |
| Modified Brighton PEWS (e)[71] | Modification of Brighton PEWS for use in a US tertiary centre. Modified wording of behaviour and respiratory effort items; altered thresholds for O₂ therapy; removed nebulisers and persistent vomiting items. No formal validation study reported. | Score | Expert opinion | No | 3 | ✓ | ✓ | ✓ | ✓ | | ✓ | ✓ | | | | ✓ | | | | | |
| Texas Children's Hospital PEWS[22] | Modification of Brighton PEWS for use in a US tertiary centre. Modified wording of behaviour category; added scoring items to respiratory and cardiovascular categories; changed O₂ therapy thresholds; modified escalation algorithm. | Score | Expert opinion | No | 5 | ✓ | ✓ | ✓ | ✓ | ✓ | ✓ | ✓ | | | | ✓ | | | | | Hourly respiratory treatments; persistent vomiting postsurgery. |
| Children's Hospital Early Warning Score[50] | Modification of Brighton PEWS for use in a US tertiary centre. Altered thresholds for O₂ therapy; changed wording for behaviour and respiratory categories; added staff and family concern; removed nebulisers and vomiting; modified escalation algorithm. | Score | Expert opinion | No | 5 | ✓ | ✓ | ✓ | ✓ | ✓ | ✓ | ✓ | | | ✓ | ✓ | | | | ✓ | |
| Children's Hospital Cardiac Early Warning Score[23 41 42 67] | Modification of Brighton PEWS for cardiac ward of a US tertiary centre. Altered O₂ therapy thresholds; added items to behaviour, respiratory and cardiovascular categories; added family and staff concern; added age-related thresholds; removed nebulisers and vomiting items; modified escalation algorithm. | Score | Expert opinion | Yes | 5 | ✓ | ✓ | ✓ | ✓ | ✓ | ✓ | ✓ | | ✓ | ✓ | ✓ | | ✓ | ✓ | ✓ | |

Continued

**Table 2** Continued

| PTTT name (references) | Development/modification details | Score/trigger | Choice of thresholds/parameters | Age-dependent thresholds | No. of items in the tool* | Respiratory rate | Heart rate | Respiratory effort/distress | LOC/behaviour | Oxygen saturation | Capillary refill time | Oxygen therapy | Systolic blood pressure | Pain | Staff concern | Skin colour | Airway problems | Temperature | Pulses | Family concern | Other items |
|---|---|---|---|---|---|---|---|---|---|---|---|---|---|---|---|---|---|---|---|---|---|
| Burn-specific PEWS[24] | Modification of Brighton PEWS, for use in a US tertiary centre. Added temperature; added intake and output scoring items; added skin component. | Score | Expert opinion | No | 6 | ✓ | ✓ | ✓ | ✓ | ✓ | ✓ | ✓ | | | | | | | ✓ | | Intake; outputs; skin. |
| Children's Hospital Los Angeles PEWS[20] | Modification of Brighton PEWS for use in a US tertiary centre. Added medical history scoring item; added single ventricle physiology scoring item; changed O₂ therapy thresholds; added items to respiratory category. | Score | Expert opinion | | 4 | ✓ | ✓ | ✓ | ✓ | ✓ | ✓ | ✓ | | | | ✓ | | | | | RRT, code blue or transfer from/to PICU in past 2 weeks; single ventricle physiology; any assisted ventilation. |
| **Melbourne Activation Criteria (MAC) and derivatives** | | | | | | | | | | | | | | | | | | | | | |
| MAC[3 13 33 62] | Initial development for use in an Australian tertiary centre to activate MET. Adapted from adult MET calling criteria, using age-appropriate thresholds. No formal validation study reported. | Trigger | Expert opinion | Yes | 9 | ✓ | ✓ | ✓ | ✓ | ✓ | ✓ | | ✓ | | ✓ | | ✓ | | | | Cardiac or respiratory arrest. |
| Modified MAC[63] | Modification of MAC for use in a Canadian tertiary centre, to activate an RRT. Removed cardiac/respiratory arrest outcome. No formal validation study reported. | Trigger | Expert opinion | Yes | 8 | ✓ | ✓ | ✓ | ✓ | ✓ | ✓ | | ✓ | | ✓ | | ✓ | | | | |
| Cardiff and Vale PEWS[32 33] | Modification of MAC for evaluation in a UK tertiary centre. Removed cardiac/respiratory arrest outcome; altered thresholds of some items; evaluated as aggregate score rather than single-item trigger. | Score | Expert opinion | Yes | 8 | ✓ | ✓ | ✓ | ✓ | ✓ | ✓ | | ✓ | | ✓ | | ✓ | | | | |
| **Bristol paediatric early warning tool (PEWT) and derivatives** | | | | | | | | | | | | | | | | | | | | | |
| Bristol PEWT[3 12 28 34 55] | Initial development for use in a UK tertiary centre. Initial candidate items drawn from unvalidated Plymouth tool—retrospectively evaluated for ability to predict adverse events among cases (n=360, HDU or PICU transfers). Development and validation dataset not independent. | Trigger | APLS values | Yes | 14 | ✓ | ✓ | ✓ | ✓ | ✓ | ✓ | ✓ | ✓ | ✓ | ✓ | | ✓ | | | | Required nebulised epinephrine; hyperkalaemia; suspected meningococcus; diabetic ketoacidosis; persistent convulsion. |
| Modified Bristol PEWT (a)[68] | Modification of Bristol PEWT for a UK tertiary centre. Adjusted wording of Airway parameters; added respiratory items; added AVPU evaluation; removed suspected meningococcus and diabetic ketoacidosis; added pH <7.2 and unresolved pain. No formal validation study reported. | Trigger | APLS values | Yes | 14 | ✓ | ✓ | ✓ | ✓ | ✓ | ✓ | ✓ | ✓ | ✓ | ✓ | | ✓ | | | | Required nebulised epinephrine or no improvement after nebulisers; pH <7.2; unresolved pain or current analgesic therapy; fitting. |

Continued

**Table 2** Continued

| PTTT name (references) | Development/ modification details | Score/ trigger | Choice of thresholds/ parameters | Age-dependent thresholds | No. of items in the tool* | PTTT parameters | | | | | | | | | | | | | | | Other items |
|---|---|---|---|---|---|---|---|---|---|---|---|---|---|---|---|---|---|---|---|---|---|
| | | | | | | Respiratory rate | Heart rate | Respiratory effort/ distress | LOC/ behaviour | Oxygen saturation | Capillary refill time | Oxygen therapy | Systolic blood pressure | Pain | Staff concern | Skin colour | Airway problems | Temperature | Pulses | Family concern | |
| Modified Bristol PEWT (b)[38] | Modification of Bristol PEWT for cardiac ward of a UK tertiary centre. Amended HR and RR thresholds. Adjusted wording of airway parameters; added respiratory items; added AVPU evaluation; removed suspected meningococcus and diabetic ketoacidosis; added pH <7.2 and unresolved pain. | Trigger | HR/RR data driven | Yes | 14 | ✓ | ✓ | ✓ | ✓ | ✓ | ✓ | ✓ | ✓ | ✓ | ✓ | | ✓ | | | | Required nebulised epinephrine or no improvement after nebulisers; pH <7.2; unresolved pain or current analgesic therapy; fitting. |

**Other PTTT**

| PTTT name (references) | Development/ modification details | Score/ trigger | Choice of thresholds/ parameters | Age-dependent thresholds | No. of items in the tool* | Respiratory rate | Heart rate | Respiratory effort/ distress | LOC/ behaviour | Oxygen saturation | Capillary refill time | Oxygen therapy | Systolic blood pressure | Pain | Staff concern | Skin colour | Airway problems | Temperature | Pulses | Family concern | Other items |
|---|---|---|---|---|---|---|---|---|---|---|---|---|---|---|---|---|---|---|---|---|---|
| NHS Institute for Innovation and Improvement PEWS[14] | Designed as part of a NHS Institute fellowship project. Adapted from adult scores and Brighton PEWS. No formal development or internal validation study published. | Score | APLS values | Yes | 6 | ✓ | ✓ | ✓ | ✓ | | | ✓ | | | ✓ | | | | | | |
| Paediatric medical emergency team (PMET) triggering criteria (a)[15] | Initial development for use in a US tertiary centre to activate a MET. Retrospective chart review of case patients (n=44, code calls) used to generate candidate items. Clinical judgement used to select final items. No formal validation of final tool reported. | Trigger | Expert opinion | No | 4 | | | ✓ | ✓ | ✓ | | | | | ✓ | ✓ | | | | ✓ | Worsening retractions; cyanosis. |
| PMET triggering criteria (b)[16] | Initial development for use in a US tertiary centre to activate a MET. Minimal description of tool development—authors deliberately chose broad criteria and categories of illness rather than specific vital signs. No formal validation study reported. | Trigger | Expert opinion | Unclear | 12 | ✓ | ✓ | ✓ | ✓ | ✓ | | | | | ✓ | | | | | ✓ | Cardiac or respiratory arrest; seizures with apnoea; progressive lethargy; circulatory compromise/acute shock syndrome. |
| Paediatric rapid response team (PRRT) triggering criteria (a)[17] | Initial development for use in a US tertiary centre, to activate an RRT. Triggering items elected through expert consensus locally—reference to similarity to MAC and PMET triggering criteria (a). No formal validation study reported. | Trigger | Expert opinion | No | 6 | ✓ | ✓ | ✓ | ✓ | ✓ | | | ✓ | | ✓ | | | | | | |
| PRRT triggering criteria (b)[18] | Initial development for use in calling RRT team in a tertiary centre in Pakistan. Minimal explanation for selection of calling criteria. No formal validation study reported in the literature. | Trigger | Unclear | Yes | 8 | ✓ | ✓ | ✓ | ✓ | ✓ | | | ✓ | | ✓ | | | | | | Convulsion. |

Continued

### Table 2 Continued

| PTTT name (references) | Development/ modification details | Score/ trigger | Choice of thresholds/ parameters | Age-dependent thresholds | No. of items in the tool* | Respiratory rate | Heart rate | Respiratory effort/ distress | LOC/ behaviour | Oxygen saturation | Capillary refill time | Oxygen therapy | Systolic blood pressure | Pain | Skin colour | Staff concern | Airway problems | Temperature | Pulses | Family concern | Other items |
|---|---|---|---|---|---|---|---|---|---|---|---|---|---|---|---|---|---|---|---|---|---|
| | | | | | | | | | | **PTTT parameters** | | | | | | | | | | | |
| Logistic regression algorithm[19] | Initial development based on data mining of electronic health records in US tertiary centre. Extracted 24 hours of clinical data from inpatients (n=6722 controls, 526 PICU transfers) and used logistic regression model to select 29 item tool. Validation performed on subset of development dataset. | Score | Expert opinion | Yes | 29 | ✓ | ✓ | ✓ | ✓ | ✓ | ✓ | ✓ | ✓ | ✓ | | | | ✓ | | | Acuity level (local measure); tissue perfusion and oxygenation. |

*Multiple parameters are often required to be collected for each scoring item/category, eg, scoring the 'cardiovascular' category in the Brighton PEWS requires collection/evaluation of HR, skin colour and capillary refill time.
†Denotes a study included in the effectiveness review.
APLS, advanced paediatric life support; AVPU, alert, voice, pain, unresponsive; HDU, high-dependency unit; HR, heart rate; LOC, level of consciousness; NHS, National Health Service; PICU, paediatric intensive care unit; PTTS, paediatric track and trigger tool; RR, respiratory rate; RRT, rapid response team.

PICU or HDU transfer[33] (AUROC=0.79, PPV=3.6%). A derivative of the MAC using an aggregate score, the Cardiff and Vale PEWS (C&VPEWS), was tested using the same cohort and outcome measures in an earlier external study (AUROC=0.86, PPV=5.9%)[32] and was the best performing PTTT in an external case-control study evaluating multiple PTTT[29] (AUROC=0.89).

### Bristol PEWT

The Bristol PEWT was evaluated by five external validation studies: two chart review studies[3 35] (no AUROC), one small cohort study of PICU transfers[34] (AUROC=0.91, PPV=11%), and two case-control studies looking at code calls[28] (AUROC=0.75) and a composite of death, arrests and PICU transfers[29] (AUROC=0.62).

### Other PTTT

The NHS III PEWS was tested by one external cohort study looking at a composite of death or unplanned transfers to PICU or HDU[14] (AUROC=0.88, PPV=4.3%) and one external case-control study looking at a composite of death, arrests and PICU transfers[29] (AUROC=0.82). Zhai et al developed and retrospectively evaluated a logistic regression algorithm in an internal case-control study looking at urgent PICU transfers in the first 24 hours of admission[19] (AUROC=0.91, PPV=4.8%).

Across PTTT, studies reporting performance characteristics of a tool at a range of different scoring thresholds demonstrate the expected interaction and trade-off between sensitivity and specificity—at lower triggering thresholds, sensitivity is high but specificity is low; at higher thresholds, the opposite is true.

### Inter-rater reliability and completeness of data

Accurate assessment of the ability of a PTTT to predict clinical deterioration is contingent on accuracy and reliability of tool scoring (whether by bedside nurses in practice or by researchers abstracting data) and the availability of underpinning observations. Only five papers made reference to accuracy or reliability of scoring,[28 31 37 42 45] with mixed results: for example, two nurses separately scoring a subset of patients on the Modified Brighton PEWS (a) achieved an intra-class coefficient of 0.92,[31] but a study nurse and bedside nurse achieved only 67% agreement in scoring the C-CHEWS tool.[37] Completeness of data was reported in 11 studies.[11 14 19 29 30 32 33 42 44 45 47] An evaluation of the Modified Bedside PEWS (a) reported that 'the PEWS was correctly performed and could be used for inclusion in the study' in 59% of cases,[30] a prospective study bench-testing the C&VPEWS found an average completeness rate of 44% for the seven different parameters in daily practice,[32] while a multicentre study of the Bedside PEWS reported that 'only 5.1% (of observation sets) had measurements on all seven items'.[44]

**Table 3** Summary of PTTT validation study outcomes

| PTTT | First author, year | Country | Study population | Study design | Number of centres | PTTT used in practice? | Internal/external validation study? | Outcome measures | Sample size | Score or trigger? | Score tested/maximum score | Which score used (frequency of scoring)* | AUROC | Sensitivity | Specificity | PPV | NPV | Notes on accuracy/reliability of scoring and missing data | Quality score (max=24) |
|---|---|---|---|---|---|---|---|---|---|---|---|---|---|---|---|---|---|---|---|
| Paediatric early warning system (PEWS) score | Duncan 2006[47] | Canada | All inpatients | Case-control study (retrospective) | 1 | No | Int | Code blue call for actual or impending cardiopulmonary arrest | 215 (87 cases) | S | 5/26 | Max 24 hours before event (hourly) | 0.90 | 78.0 | 95.0 | 4.2† | | No details on data abstraction. 13% of eligible cases and 84% of eligible controls excluded due to incomplete clinical data. | 14 |
| | Robson 2013[28] | USA | All inpatients | Case-control study (retrospective) | 1 | No | Ext | Code blue call | 192 (96 cases) | S | 5/32 | Max 24 hours before event (six hourly) | 0.85 | 86.6 | 72.2 | | | Four researchers scored PTTT from 20 charts, inter-rater reliability of 0.95. No details on extent of missing data. | 8 |
| | Chapman 2017[29] | UK | All inpatients | Case-control study (retrospective) | 1 | No | Ext | Death, arrest or unplanned PICU transfer | 608 (297 cases) | S | 7/32 | Max 48 hours before event (per usual practice) | 0.82 | 70.0 | 75.0 | 72.6 | 72.0 | Data abstraction by single researcher, 36% of observation sets contained HR, RR, O2 Sats, systolic BP, temperature and assessment of consciousness. | 17 |
| Bedside PEWS | Parshuram 2009[11] | Canada | All inpatients | Case-control study (retrospective) | 1 | No | Int | Urgent PICU transfer (without code blue call) | 180 (60 cases) | S | 8/26 | Max 24 hours before event (hourly) | 0.91 | 82.0 | 93.0 | | | Availability of scoring items in medical records varied from 27% (cap refill time) to 93% (oxygen therapy). | 21 |
| | Parshuram 2011[44] | Canada and UK | All inpatients | Case-control study (prospective) | 4 | No | Ext | Urgent PICU transfer or immediate call to resuscitation team | 2074 (686 cases) | S | 7/26 | Max 24 hours before event (hourly) | 0.87 | 64.0 | 91.0 | | | PTTT scores calculated electronically after abstraction by research nurse. 5.1% of records had all seven items recorded, 31% had at least five items. | 22 |
| | Robson 2013[28] | USA | All inpatients | Case-control study (retrospective) | 1 | No | Ext | Code blue call | 192 (96 cases) | S | 7/26 | Max 24 hours before event (six hourly) | 0.73 | 56.3 | 78.1 | | | See above. | 8 |
| | Zhai 2014[19] | USA | All inpatients | Case-control study (retrospective) | 1 | No | Ext | Urgent PCU transfer within 24 hours of admission | 6352 (53 cases) | S | 7/26 | Max 24 hours before event (hourly) | 0.82 | 73.6 | 71.7 | 2.1† | | Data extracted from electronic health records. Excluded two items of Bedside PEWS (oxygen therapy and respiratory effort) due to difficulty abstracting. | 17 |
| | Gawronski 2016[46] | Italy | Stem Cell Transplant Unit | Case-control study (retrospective) | 1 | No | Ext | Unexpected death, urgent consult with RRT or urgent PICU transfer | 99 (19 cases) | S | 6/26 | Score 4 hours before event | 0.90 | 79.0 | 97.5 | | | Data abstracted by research nurses. No details on extent of missing data. Conflicting/missing observations resolved by interviews with clinical staff. | 15 |
| | Chapman 2017[29] | UK | All inpatients | Case-control study (retrospective) | 1 | No | Ext | Death, arrest or PICU transfer | 608 (297 cases) | S | 6/26 | Max 48 hours before event (per usual practice) | 0.88 | 72.0 | 89.0 | 86.0 | 77.0 | See above. | 17 |
| Modified Bedside PEWS (a) | Fujikschot 2015[30] (study 1) | The Netherlands | Oncology ward | Case-cohort study (retrospective) | 1 | Yes | Int | Emergency medical intervention or reviewed by PICU staff or staff concern | 118 (15 cases) | S | 8/28 | Unclear (minimum eight hourly) | | | | 73.0 | | 41% of admissions excluded from study due to incomplete PTTT scores. | 10 |
| | Fujikschot 2015[30] (study 2) | The Netherlands | All inpatients | Case-cohort study (retrospective) | 1 | Yes | Int | PICU transfer | Unclear (24 cases) | S | 8/28 | Score 2-6 hours before event (minimum eight hourly) | | 66.6 | | | | High rate of exclusions reported due to missing data. | 10 |
| | Fujikschot 2015[30] (study 3) | The Netherlands | All inpatients | Case-cohort study (prospective) | 1 | Yes | Int | Emergency medical intervention | Unclear (14 cases) | S | 8/28 | Unclear (minimum eight hourly) | | 100 | | | | No details on missing data. | 10 |
| | Chapman 2017[29] | UK | All inpatients | Case-control study (retrospective) | 1 | No | Ext | Death, arrest or PICU transfer | 608 (297 cases) | S | 7/28 | Max 48 hours before event (per usual practice) | 0.87 | 69.0 | 91.0 | 87.9 | 79.0 | See above. | 17 |
| Modified Bedside PEWS (b) | Ross 2015[48] | USA | All inpatients | Case-control study (retrospective) | 1 | No | Int | Urgent PICU transfer | 4628 (848 cases) | S | 8/26 | Max during admission | | 70.0 | 84.0 | | | No details on data abstraction. Respiratory effort category excluded due to difficulty abstracting. No details on missing data. | 9 |

Continued

**Table 3** Continued

| PTTT | First author, year | Country | Study population | Study design | Number of centres | PTTT used in practice? | Internal/external validation study? | Outcome measures | Sample size | Score or trigger? | Score tested/maximum score | Which score used (frequency of scoring)* | AUROC | Sensitivity | Specificity | PPV | NPV | Notes on accuracy/reliability of scoring and missing data | Quality score (max=24) |
|---|---|---|---|---|---|---|---|---|---|---|---|---|---|---|---|---|---|---|---|
| Modified Brighton PEWS (a) | Tucker 2009[31] | USA | General medical unit | Cohort study (prospective) | 1 | Yes | Int | PICU transfer | 2979 (51 cases) | S | 3/11 | Max during admission (four hourly) | 0.89 | 90.2 | 74.4 | 5.8 | 99.8 | Intraclass coefficient of 0.92 reported for two bedside nurses scoring 55 patients. No details on missing data. | 14 |
| | Zhai 2014[19] | USA | All inpatients | Case-control study (retrospective) | 1 | No | Ext | Urgent PICU transfer within 24hours of admission | 6352 (53 cases) | S | 2/11 | Max 24 hours before event (hourly) | 0.74 | 68.4 | 81.6 | 2.3 | | Data extracted from electronic health records. Only included records with complete PEWS score: 64% of eligible cases and 51% of eligible controls excluded. | 17 |
| | Fenix 2015[39] | USA | PICU transfers among all inpatients (excluding haematology, oncology, surgical and cardiac wards) | Case-control study (retrospective) | 1 | Yes | Ext | Non-elective PICU transfer followed by deterioration event | 97 PICU transfers (51 cases of PICU transfer followed by 'deterioration event') | S | 3/11 | Max during admission | | 80.0 | 43.0 | 61.0 | 67.0 | No details on missing data. | 15 |
| Modified Brighton PEWS (b) | Akre 2010[45] | USA | All inpatients | Chart review study (retrospective) | 1 | No | Int | Rapid response team call or code blue call | 186 cases (170 RRT calls, 16 code calls) | S | 4/13 | Max 24 hours before event (minimum four hourly) | | 85.5 | | | | Scores abstracted from charts by single nurse, having calibrated with advanced nurse practitioner. Categories scored missing if any items missing. 25% of charts missing behavioural state, 26% cardiovascular colour. | 14 |
| | Chapman 2017[29] | UK | All inpatients | Case-control study (retrospective) | 1 | No | Ext | Death, arrest or PICU transfer | 608 (297 cases) | S | 4/13 | Max 48 hours before event (per usual practice) | 0.79 | 61.0 | 84.0 | 78.4 | 69.0 | See above. | 17 |
| Modified Brighton PEWS (d) | Skaletzky 2012[48] | USA | Medical surgical wards | Case-control study (retrospective) | 1 | No | Int | PICU transfer | 350 (100 cases) | S | 2.5/9 | Max 48 hours before event (four hourly) | 0.81 | 62.0 | 89.0 | | | Data abstracted from medial charts and notes. Behaviour category abstracted from LOC. No details on missing data. | 15 |
| | Chapman 2017[29] | UK | All inpatients | Case-control study (retrospective) | 1 | No | Ext | Death, arrest or PICU transfer | 608 (297 cases) | S | 4/9 | Max 48 hours before event (per usual practice) | 0.74 | 46.0 | 90.0 | 81.3 | 63.0 | See above. | 17 |
| Children's Hospital Early Warning Score | McLellan 2014[50] | USA | All inpatients | Case-control study (retrospective) | 1 | Yes | Int | Arrest or unplanned PICU transfer | 1136 (360 cases) | S | 4/12 | Max in admission (four hourly) | 0.90 | 84.2 | 80.9 | | | No details on missing data. | 10 |
| | McLellan 2013[23] | USA | Cardiovascular unit | Case-control study (retrospective) | 1 | Yes | Int | Arrest or unplanned PICU transfer | 312 (64 cases) | S | 3/12 | Max 18 hours before event (four hourly) | 0.86 | 95.3 | 76.2 | 50.8 | 98.4 | Study nurse and bedside nurses assessed scores for 37 patients. 67% agreement. No details on missing data. | 9 |
| | Agulnik 2016[41] | USA | Oncology unit | Case-control study (retrospective) | 1 | Yes | Ext | Unplanned PICU transfer | 330 (110 cases) | S | 4/12 | Max 24 hours before event (four hourly) | 0.96 | 86.0 | 95.0 | | | PTTT scores abstracted by researcher. Did not abstract if vital signs were present but no PTTT score calculated by nurse. No details on missing data. | 14 |
| | Agulnik 2017[42] | Guatemala | Oncology unit | Case-control study (retrospective) | 1 | Yes | Ext | Unplanned PICU transfer | 258 (129 cases) | S | 4/12 | Max 24 hours before event (three hourly) | | 91.0 | 88.0 | | | Researcher evaluated charts and calculated scores, reporting 14% error rate (PTTT score calculated incorrectly) and 3% omission rate (vital signs recorded but no PTTT score calculated). One out of 130 cases excluded due to missing PTTT documentation. | 16 |
| Children's Hospital Los Angeles PEWS | Mandell 2015[40] | USA | Inpatients discharged from PICU to ward | Case-control study (retrospective) | 1 | Yes | Int | Early unplanned re-admission to PICU (within 48hours of discharge from PICU to ward) | 189 (38 cases) | S | 2/10 | First score assigned on ward, post-PICU discharge | 0.71 | 76.0 | 56.0 | | | No details on missing data. | 12 |

Continued

**Table 3** Continued

| PTTT | First author, year | Country | Study population | Study design | Number of centres | PTTT used in practice? | Internal/external validation study? | Outcome measures | Sample size | Score or trigger? | Score tested/maximum score | Which score used (frequency of scoring)? | AUROC | Sensitivity | Specificity | PPV | NPV | Notes on accuracy/reliability of scoring and missing data | Quality score (max=24) |
|---|---|---|---|---|---|---|---|---|---|---|---|---|---|---|---|---|---|---|---|
| Melbourne Activation Criteria | Tume 2007[9] | UK | Inpatients with an unplanned PICU transfer | Chart review study (retrospective) | 1 | No | Ext | Unplanned PICU transfer | 33 cases | T | N/A | Unclear | | 87.8 | | | | Data abstracted by two reviewers. Reference to 'large number of missing records and observation charts'. | 11 |
| | Tume 2007[9] | UK | Inpatients with an unplanned PHDU transfer | Chart review study (retrospective) | 1 | No | Ext | Unplanned PHDU transfer | 32 cases | T | N/A | Unclear | | 87.5 | | | | See above. | 11 |
| | Edwards 2011[33] | UK | All inpatients | Cohort study (retrospective) | 1 | No | Ext | Death or unplanned PICU or HDU transfer | 1000 (16 cases) | T | N/A | Any trigger over admission (per usual practice) | 0.79 | 68.3 | 83.2 | 3.6 | 99.7 | Observation charts altered to include all PTTT parameters. 56% of records missing at least one component. Missing data assumed to be normal. | 17 |
| | Chapman 2017[29] | UK | All inpatients | Case-control study (retrospective) | 1 | No | Ext | Death, arrest or PICU transfer | 608 (297 cases) | T | N/A | Max 48 hours before event (per usual practice) | 0.71 | 93.0 | 49.0 | 64.0 | 88.0 | See above. | 17 |
| Cardiff and Vale PEWS | Edwards 2009[32] | UK | All inpatients | Cohort study (prospective) | 1 | No | Int | Death or unplanned PICU or HDU transfer | 1000 (16 cases) | S | 2/8 | Max score during admission (per usual practice) | 0.86 | 69.5 | 89.9 | 5.9 | 99.7 | Observation charts altered to include all PTTT parameters. 56% of records missing at least one component. Missing data assumed to be normal. | 18 |
| | Chapman 2017[29] | UK | All inpatients | Case-control study (retrospective) | 1 | No | Ext | Death, arrest or PICU transfer | 608 (297 cases) | S | 3/8 | Max 48 hours before event (per usual practice) | 0.89 | 80.0 | 86.0 | 84.0 | 82.0 | See above. | 17 |
| Bristol paediatric early warning tool (PEWT) | Tume 2007[9] | UK | Inpatients with an unplanned PICU transfer | Chart review (retrospective) | 1 | No | Ext | Unplanned PICU transfer | 33 cases | T | N/A | Unclear | | 87.8 | | | | See above. | 11 |
| | Tume 2007[9] | UK | Inpatients with an unplanned PHDU transfer | Chart review (retrospective) | 1 | No | Ext | Unplanned PHDU transfer | 32 cases | T | N/A | Unclear | | 84.4 | | | | See above. | 11 |
| | Wright 2011[35] | UK | All inpatients | Chart review (retrospective) | 1 | Yes | Ext | Cardiac arrest | 55 cases | T | N/A | If triggered 24 hours before event | 0.91 | 49.1 | | | | One case excluded due to missing notes. No details on missing data. | 11 |
| | O'Loughlin 2012[34] | UK | All inpatients | Cohort study (prospective) | 1 | Yes | Ext | PICU transfer | 331 (7 cases) | T | N/A | Triggered during admission (12 hourly) | | 100 | 81.0 | 11.0 | | No details on missing data. | 6 |
| | Robson 2013[28] | USA | All inpatients | Case-control study (retrospective) | 1 | No | Ext | Code blue call | 192 (96 cases) | T | N/A | Triggered 24 hours before event (6 hourly) | 0.75 | 76.3 | 61.5 | | | See above. | 8 |
| | Chapman 2017[29] | UK | All inpatients | Case-control study (retrospective) | 1 | No | Ext | Death, arrest or PICU transfer | 608 (297 cases) | T | N/A | If triggered 48 hours before event (per usual practice) | 0.62 | 96.0 | 28.0 | 56.0 | 88.0 | See above. | 17 |
| Modified Bristol PEWT (b) | Clayson 2014[36] | UK | Cardiac ward | Cohort study (prospective) | 1 | Yes | Int | 'A deteriorating patient' | 126 (unclear number of cases) | T | N/A | Unclear | | | | 12.5 | 97.0 | No details on missing data. | 5 |
| NHS Institute for Innovation and Improvement PEWS | Mason 2016[14] | UK | All inpatients | Cohort study (retrospective) | 1 | No | Ext | Death or unplanned PICU or HDU transfer | 1000 (16 cases) | S | 2/7 | Max score over admission (per usual practice) | 0.88 | 80.0 | 81.0 | 4.3 | 99.7 | Observation charts altered to include all PTTT parameters. 56% of records missing at least one component. Missing data assumed to be normal. | 15 |
| | Chapman 2017[29] | UK | All inpatients | Case-control study (retrospective) | 1 | No | Ext | Death, arrest or PICU transfer | 608 (297 cases) | S | 2/7 | Max 48 hours before event (per usual practice) | 0.82 | 83.0 | 65.0 | 69.6 | 80.0 | See above. | 17 |
| Logistic regression algorithm | Zhai 2014[19] | USA | All inpatients | Case-control study (retrospective) | 1 | No | Ext | Urgent PICU transfer within 24 hours of admission | 6352 (53 cases) | S | >0.5 | Max 24 hours before event (hourly) | 0.91 | 84.9 | 85.9 | 4.8 | 80.0 | Data extracted from electronic health records. No details on extent of missing data but authors report that 'missing data were a major cause of incorrect prediction'. | 17 |

Continued

## Table 3 Continued

| PTTT | First author, year | Country | Study population | Study design | Number of centres | PTTT used in practice? | Internal/ external validation study? | Outcome measures | Sample size | Score or trigger? | Score tested/ maximum score | Which score used (frequency of scoring)?* | AUROC | Sensitivity | Specificity | PPV | NPV | Notes on accuracy/ reliability of scoring and missing data | Quality score (max=24) |
|---|---|---|---|---|---|---|---|---|---|---|---|---|---|---|---|---|---|---|---|
| Burton Paediatric Early Warning Score | Ahmed 2012[36] | UK | PICU admissions only | Chart review (retrospective) | 1 | Yes | Int | PICU admission | 23 | S | 4/19 | Max 24 hours before event (Unclear) | | 93.0 | | | | Data extracted from case notes by two reviewers. No details on missing data. | 4 |
| 'Between the Flags' PEWS | Blackstone 2017[43] | UK | Urgent PICU admissions only | Chart review (retrospective) | 1 | Yes | Ext | Urgent PICU admission | 100 | T | N/A | Unclear | | 91.0 | | | | Data extracted from health records. No details on missing data. | 8 |

All studies conducted in a specialist/tertiary centre.

PPV and NPV values in italics represent results from case-control studies—these values are misleading in isolation because they assume that the wider prevalence rate of the adverse event is equal to the case to control ratio used in the research study (eg, if the researchers studied 300 cases and 300 controls, the prevalence rate of adverse events for the calculation of PPV is 50%). As per the cohort studies, prevalence rates of critical events are typically far lower among hospitalised paediatric populations than the case-control ratios used in studies, and so PPV values would be considerably lower in clinical practice.

Studies classified as internal validation if the setting for the study was the same hospital and same research team as those who developed the study. Studies classified as external validation if the score was tested in a different centre and by a different research team to those who developed it.

*Typically, study researchers collected or abstracted multiple PTTT scores for each patient at different time points, but can only use one score per patient for the analysis of the tool's predictive ability. This column specifies which score the researchers used. In most cases, the study team used the maximum PTTT score recorded for each patient in a given study window, eg, 24 hours prior to a critical event for case patients. The text in parentheses describes the frequency with which scores were assessed or abstracted for each patient, if this information was described in the paper.

†Case-control study, but PPV value calculated based on clinical prevalence of event as measured at local centre during the study.

AUROC, area under the receiver operating characteristic curve; Ext, external validation; HFNC, high flow nasal cannula; Int, internal validation; Max, maximum; N/A, not applicable; NPV, negative predictive value; PHDU, paediatric high-dependency unit; PICU, paediatric intensive care unit; PPV, positive predictive value; PTTT, paediatric track and trigger tool; RRT, rapid response team; S, score; T, trigger.

## Question 2: how effective are early warning systems at reducing mortality and critical events in hospitalised children?

Eleven papers meeting inclusion criteria were excluded from analysis for providing insufficient statistical information (eg, denominator data, absolute numbers of events) to calculate effect sizes.[39 51–59] Further details on papers excluded from analysis are provided in online supplementary table 5. Findings from the 19 studies included in the analysis are summarised in table 4.

### Type of early warning system interventions

Seventeen interventions involved the introduction of a new PTTT,[13 15–18 60–72] one intervention introduced a mandatory triggering element to an existing PTTT[71] and one study reported a large, multicentre analysis of MET introduction with no details on PTTT use.[73] Twelve interventions included the introduction of a new MET or RRT,[13 15–18 60–65 69] while four further interventions introduced a new PTTT in a hospital with an existing MET or RRT. Only three studies therefore evaluated a PTTT in the absence of a dedicated response team.[67 68 70] A staff education programme was explicitly described in 10 interventions.[13 15 17 61 62 64 67 68 70 72]

Of the 18 studies that used a PTTT, only 7 used a tool that had been formally evaluated for validity: 3 used the Bedside PEWS,[64 65 70] 2 used the MAC,[13 62] 1 used the Modified Brighton PEWS (b)[72] and 1 used the C-CHEWS.[67] One study did not report the PTTT used,[61] while 10 studies used a variety of calling criteria and local modifications to validated tools that had not been evaluated for validity.[15–18 60 63 66 68 69 71]

### Mortality (ward or hospital wide)

Two uncontrolled before-after studies (both with MET/RRT) reported significant mortality rate reductions postintervention: one in hospital wide deaths per 100 discharges[17] (RR=0.82, 95% CI 0.70 to 0.95) and one in total hospital deaths per 1000 admissions (RR=0.65, 95% CI 0.57 to 0.75) and deaths on the ward ('unexpected deaths') per 1000 admissions[62] (RR=0.35, 95% CI 0.13 to 0.92). Seven studies found no reductions in mortality, including two high-quality multicentre studies.[13 15 60 63–65 73] Parshuram *et al* conducted a cluster randomised trial and found no difference in all-cause hospital mortality rates between 10 hospitals randomly selected to receive an intervention centred around use of the Bedside PEWS and 11 usual care hospitals, 1-year postintervention (OR=1.01, 95% CI 0.61 to 1.69).[64] Kutty *et al*[73] assessed the impact of MET implementation in 38 US paediatric hospitals with an interrupted time series study, and reported no difference in the slope of hospital mortality rates 5 years postintervention and the expected slope based on preimplementation trends (OR=0.94, 95% CI 0.93 to 0.95).

### PICU mortality

Two uncontrolled before-after studies (both with MET/RRT) reported a significant postintervention reduction in

**Table 4** Summary of early warning system effectiveness study outcomes

| Outcome | First author, year | Implemented a new PTTT | Implemented new RRT/MET | Modified escalation process | Staff training/education | PTTT | Number of centres | Specialist unit? | Existing RRT/MET? | Population | Study design | Study duration in months | Events before, n (rate) | Events after, n (rate) | Effect size (95% CI) | P value | Quality score (max=26) |
|---|---|---|---|---|---|---|---|---|---|---|---|---|---|---|---|---|---|
| **Mortality** | | | | | | | | | | | | | | | | | |
| Deaths on ward (per 1000 admissions) | Tibballs 2005[13] | ✓ | | | ✓ | Melbourne Activation Criteria | 1 | Y | N | All inpatients | Uncontrolled before-after study (prospective) | 53 (41 before, 12 after) | 13 (0.12) | 2 (0.06) | RR=0.45 (0.10 to 1.99)† | 0.29 | 10 |
| Hospital-wide deaths (per 100 discharges) | Sharek 2007[17] | ✓ | ✓ | | ✓ | Paediatric rapid response team (RRT) triggering criteria | 1 | Y | N | All inpatients | Uncontrolled before-after study (prospective) | 84 (67 before, 17 after) | 547 (1.01) | 158 (0.83) | RR=0.82 (0.70 to 0.95) | **0.007** | 15 |
| Hospital-wide deaths, excluding neonate ICU and ED (per 1000 discharges) | Zenker 2007[60] | ✓ | ✓ | | | RRT activation criteria* | 1 | Y | N | All inpatients | Uncontrolled before-after study (prospective) | 34 (23 before, 11 after) | 97 (4.30) | 52 (4.45) | RR=1.04 (0.74 to 1.45)† | 0.57 | 12 |
| Deaths outside ICU (per 1000 non-ICU patient-days) | Brilli 2007[15] | ✓ | ✓ | | ✓ | Paediatric medical emergency team triggering criteria (a) | 1 | Y | N | All inpatients | Uncontrolled before-after study (prospective) | 27 (15 before, 12 after) | 9 (0.10) | 2 (0.04) | RR=0.39 (0.08 to 1.80)† | 0.13 | 14 |
| Ward death rate (per 1000 ward admissions) | Hanson 2010[61] | ✓ | ✓ | | ✓ | Not described | 1 | Y | N | All inpatients | Uncontrolled before-after study (retrospective) | 36 (24 before, 12 after) | 13 (1.50) | 2 (0.45) | RR=0.30 (0.07 to 1.31)† | 0.07 | 18 |
| Total hospital deaths (per 1000 admissions) | Tibballs 2009[62] | ✓ | ✓ | | ✓ | Melbourne Activation Criteria | 1 | Y | N | All inpatients | Uncontrolled before-after study (prospective) | 89 (41 before, 48 after) | 459 (4.38) | 398 (2.87) | RR=0.65 (0.57 to 0.75) | <0.0001 | 15 |
| Deaths on ward (per 1000 admissions) | Tibballs 2009[62] | ✓ | ✓ | | ✓ | Melbourne Activation Criteria | 1 | Y | N | All inpatients | Uncontrolled before-after study (prospective) | 89 (41 before, 48 after) | 13 (0.12) | 6 (0.04) | RR=0.35 (0.13 to 0.92) | **0.03** | 15 |
| All-cause hospital mortality (per 1000 admissions) | Kotsakis 2011[63] | ✓ | ✓ | | ✓ | Modified Melbourne Activation Criteria | 4 | Y | N | All inpatients | Uncontrolled before-after study (prospective) | 48 (24 before, 24 after) | 553 (9.97) | 540 (9.65) | RR=0.97 (0.83 to 1.12) | 0.65 | 18 |
| All-cause hospital mortality (per 1000 discharges) | Parshuram 2018[64] | ✓ | ✓ | | ✓ | Bedside PEWS | 21 | Y | N | All inpatients | Cluster randomised trial (prospective) | 18 (6 pre, 12 post) | Con: 61 (1.31)<br>Int: 52 (1.95) | Con: 147 (1.56)<br>Int: 97 (1.93) | OR=1.01 (0.61 to 1.69) | 0.96 | 23 |
| Hospital mortality (per 1000 admissions) | Kutty 2018[73] | ✓ | | | | NR | 38 | Y | N | All inpatients | Interrupted time series (retrospective) | 180 (60 before, 120 after) | N/A | N/A | OR=0.94 (0.93 to 0.95) | 0.98 | 20 |
| **PICU mortality** | | | | | | | | | | | | | | | | | |
| PICU mortality after PICU admission from ward (per PICU admission) | Anwar-ul-Haque, 2010[18] | ✓ | ✓ | | | Paediatric RRT triggering criteria (b) | 1 | Y | N | All inpatients | Uncontrolled before-after study (retrospective) | 18 (9 before, 9 after) | 23 (51.11) | 5 (15.63) | RR=0.31 (0.13 to 0.72)† | **0.007**† | 6 |
| PICU mortality after PICU readmission within 48 hours of discharge (per 1000 admissions) | Kotsakis 2011[63] | ✓ | ✓ | | | Modified Melbourne Activation Criteria | 4 | Y | N | All inpatients | Uncontrolled before-after study (retrospective) | 48 (24 before, 24 after) | 16 (0.29) | 7 (0.13) | RR=0.43 (0.17 to 0.99) | <0.05 | 18 |
| PICU mortality after urgent PICU admission from ward (per 1000 admissions) | Kotsakis 2011[63] | ✓ | ✓ | | ✓ | Modified Melbourne Activation Criteria | 4 | Y | N | All inpatients | Uncontrolled before-after study (retrospective) | 48 (24 before, 24 after) | 70 (1.3) | 61 (1.1) | RR=0.90 (0.70 to 1.00) | 0.25 | 18 |
| Death prior to discharge (per unplanned PICU transfer) | Bonafide 2014[65] | ✓ | ✓ | | | Bedside PEWS | 1 | Y | N | All inpatients | Interrupted time series study (retrospective) | 59 (32 before, 27 after) | 51 (6.3) | 56 (6.5) | RR=1.03 (0.72 to 1.49)† | 0.99 | 23 |
| PICU mortality (per PICU admission) | Duns 2014[66] | ✓ | ✓ | | | Between the Flags (BTS) tool* | 1 | Y | Y | All inpatients | Uncontrolled before-after study (prospective) | 48 (24 before, 24 after) | 30 (8.57) | 20 (5.49) | RR=0.64 (0.37 to 1.11)† | 0.14 | 7 |
| Death in PICU (per 1000 patient-days) | Agulnik 2017[67] | ✓ | | | ✓ | Children's Hospital Cardiac Early Warning Score (C-CHEWS) | 1 | Y | N | Oncology unit | Uncontrolled before-after study (prospective) | 24 (12 before, 12 after) | 21 (1.25) | 22 (1.10) | RR=0.89 (0.49 to 1.61)† | 0.76 | 19 |
| Death in PICU (per emergency PICU admission) | Sefton 2015[68] | ✓ | | ✓ | ✓ | Modified Bristol PEWT (a) | 1 | Y | N | All PICU admissions | Controlled before-after study (retrospective) | 24 (12 before, 12 after) | 17 (10.8) | 14 (8.4) | RR=0.78 (0.40 to 1.53)† | 0.47 | 16 |
| Deaths in PICU (per unplanned PICU admission) | Kolovos 2018[69] | ✓ | ✓ | | | RRT activation criteria* | 1 | Y | N | All unplanned PICU admissions | Uncontrolled before-after study (retrospective) | 78 (42 before, 36 after) | 54† (4.9) | 40† (3.8) | RR=0.77 (0.52 to 1.15)† | 0.20† | 12 |
| PICU mortality (per 1000 discharges) | Parshuram 2018[64] | ✓ | ✓ | | ✓ | Bedside PEWS | 21 | Y | N | All inpatients | Cluster randomised trial (prospective) | 18 (6 pre, 12 post) | Con: 34 (0.73)<br>Int: 33 (1.24) | Con: 91 (0.96)<br>Int: 56 (1.12) | OR=0.95 (0.48 to 1.86) | 0.88 | 23 |
| **Cardiac arrest** | | | | | | | | | | | | | | | | | |
| Cardiac arrests on ward (per 1000 admissions) | Tibballs 2005[13] | ✓ | ✓ | | ✓ | Melbourne Activation Criteria | 1 | Y | N | All inpatients | Uncontrolled before-after study (prospective) | 53 (41 before, 12 after) | 20 (0.19) | 4 (0.11) | RR=0.58 (0.20 to 1.70) | 0.33 | 10 |
| Cardiopulmonary arrests (per 1000 non-ICU patient-days) | Brilli 2007[15] | ✓ | ✓ | | ✓ | Paediatric medical emergency team triggering criteria (a) | 1 | Y | N | All inpatients | Uncontrolled before-after study (prospective) | 27 (15 before, 12 after) | 7 (0.08) | 2 (0.04) | RR=0.50 (0.10 to 2.42)† | 0.11 | 14 |

Continued

Note: Parshuram 2018[64] study country — Belgium, The Netherlands, Ireland, Italy, Canada, New Zealand.

**Table 4** Continued

| Outcome | First author, year | Intervention — Implemented a new PTTT | Intervention — Implemented new RRT/MET | Intervention — Modified escalation process | Intervention — Staff training/education | PTTT | Country | Number of centres | Specialist unit? | Existing RRT/MET? | Population | Study design | Study duration in months | Events before, n (rate) | Events after, n (rate) | Effect size (95% CI) | P value | Quality score (max=26) |
|---|---|---|---|---|---|---|---|---|---|---|---|---|---|---|---|---|---|---|
| Ward cardiac arrest rate (per 1000 ward admissions) | Hanson 2010[61] | ✓ | ✓ | | ✓ | Not described | USA | 1 | Y | N | All inpatients | Uncontrolled before-after study (retrospective) | 36 (24 before, 12 after) | 11 (1.27) | 2 (0.45) | RR=0.35 (0.08 to 1.58)† | 0.13 | 18 |
| Ward cardiopulmonary arrests (per 1000 patient-days) | Hunt 2008[16] | ✓ | ✓ | | | Paediatric medical emergency team triggering criteria | USA | 1 | Y | N | All inpatients | Uncontrolled before-after study (prospective) | 24 (12 before, 12 after) | 5 (0.10) | 5 (0.10) | RR=0.98 (0.22 to 4.24) | 0.97 | 17 |
| Preventable cardiac arrests (per 1000 admissions) | Tibballs 2009[62] | ✓ | ✓ | | ✓ | Melbourne Activation Criteria | Australia | 1 | Y | N | All inpatients | Uncontrolled before-after study (prospective) | 89 (41 before, 48 after) | 17 (0.16) | 10 (0.07) | RR=0.45 (0.20 to 0.97) | **0.04** | 15 |
| Unexpected cardiac arrests (per 1000 admissions) | Tibballs 2009[62] | ✓ | ✓ | | ✓ | Melbourne Activation Criteria | Australia | 1 | Y | N | All inpatients | Uncontrolled before-after study (prospective) | 89 (41 before, 48 after) | 20 (0.19) | 24 (0.17) | RR=0.91 (0.50 to 1.64) | 0.75 | 15 |
| Actual cardiopulmonary arrests (per 1000 ward admissions) | Kotsakis 2011[63] | ✓ | ✓ | | | Modified Melbourne Activation Criteria | Canada | 4 | Y | N | All inpatients | Uncontrolled before-after study (prospective) | 48 (24 before, 24 after) | 69 (1.9) | 66 (1.8) | RR=0.96 (0.76 to 1.96) | 0.68 | 18 |
| Near cardiopulmonary arrests (per 1000 admissions) | Kotsakis 2011[63] | ✓ | ✓ | | | Modified Melbourne Activation Criteria | Canada | 4 | Y | N | All inpatients | Uncontrolled before-after study (prospective) | 48 (24 before, 24 after) | 123 (3.4) | 67 (1.9) | RR=0.54 (0.52 to 0.57) | <0.001 | 18 |
| Cardiac arrests on ward (per 1000 non-ICU patient-days) | Bonafide 2014[65] | ✓ | ✓ | | | Bedside PEWS | USA | 1 | Y | N | All inpatients | Interrupted time series study (prospective) | 59 (32 before, 27 after) | 6† (0.03) | 2† (0.01) | RR=0.36 (0.07 to 1.78)† | 0.21 | 23 |
| Cardiac arrests (per 1000 patient-days) | Parshuram 2018[64] | ✓ | ✓ | | ✓ | Bedside PEWS | Belgium, Ireland, The Netherlands, England, Italy, Canada, New Zealand | 21 | Y | N | All inpatients | Cluster randomised trial (prospective) | 18 (six pre, 12 post) | Con: 18 (0.11)  Int: 15 (0.12) | Con: 32 (0.10)  Int: 27 (0.11) | RR=1.02 (0.65 to 1.62) | 0.92 | 23 |
| **Respiratory arrest** | | | | | | | | | | | | | | | | | | |
| Ward respiratory arrests (per 1000 patient-days) | Hunt 2008[16] | ✓ | ✓ | | | Paediatric medical emergency team triggering criteria | USA | 1 | Y | N | All inpatients | Uncontrolled before-after study (prospective) | 24 (12 before, 12 after) | 11 (0.23) | 3 (0.06) | RR=0.27 (0.07 to 0.95) | **0.04** | 17 |
| **Cardiac or respiratory arrest** | | | | | | | | | | | | | | | | | | |
| Cardiac or respiratory arrest (per 1000 discharges) | Zenker 2007[60] | ✓ | ✓ | | | RRT activation criteria* | USA | 1 | Y | N | All inpatients | Uncontrolled before-after study (prospective) | 34 (23 before, 11 after) | 180 (7.98) | 60 (5.13) | RR=0.64 (0.48 to 0.86)† | 0.19 | 12 |
| Code calls (per 1000 non-ICU patient-days) | Brilli 2007[15] | ✓ | ✓ | | ✓ | Paediatric medical emergency team triggering criteria (a) | USA | 1 | Y | N | All inpatients | Uncontrolled before-after study (prospective) | 27 (15 before, 12 after) | 25 (0.27) | 6 (0.11) | RR=0.42 (0.17 to 1.03)† | 0.06† | 14 |
| Code calls (per 1000 non-ICU patient-days) | Sharek 2007[17] | ✓ | ✓ | | ✓ | Paediatric RRT triggering criteria | USA | 1 | Y | N | All inpatients | Uncontrolled before-after study (prospective) | 84 (67 before, 17 after) | 53 (0.52) | 5 (0.15) | RR=0.29 (0.10 to 0.65) | **0.008** | 15 |
| Code calls (per 1000 admissions) | Anwar-al-Haque 2010[18] | ✓ | ✓ | | | Paediatric RRT triggering criteria (b) | Pakistan | 1 | Y | N | All inpatients | Uncontrolled before-after study (retrospective) | 18 (9 before, 9 after) | 26 (5.25) | 12 (2.73) | RR=0.52 (0.26 to 1.03) | 0.06 | 6 |
| **Calls for urgent review/assistance** | | | | | | | | | | | | | | | | | | |
| Urgent calls to respiratory therapist (per 1000 patient-days) | Parshuram 2011[70] | ✓ | | ✓ | ✓ | Bedside PEWS | Canada | 1 | N | N | All inpatients | Uncontrolled before-after study (prospective) | 8 (3 before, 5 after) | 8 (9.5) | 8 (3.4) | RR=0.36 (0.13 to 0.95)† | **0.04†** | 23 |
| Urgent calls to paediatrician (per 1000 patient-days) | Parshuram 2011[70] | ✓ | | ✓ | ✓ | Bedside PEWS | Canada | 1 | N | N | All inpatients | Uncontrolled before-after study (prospective) | 8 (3 before, 5 after) | 19 (22.6) | 12 (5.1) | RR=0.23 (0.11 to 0.46)† | <0.0001 | 23 |
| Code blue calls on the ward (per 1000 admissions) | Kotsakis 2011[63] | ✓ | ✓ | | ✓ | Modified Melbourne Activation Criteria | Canada | 4 | Y | N | All inpatients | Uncontrolled before-after study (prospective) | 48 (24 before, 24 after) | 210 (3.75) | 150 (2.70) | RR=0.71 (0.61 to 0.83) | <0.0001 | 18 |
| Urgent calls to outreach team (per 1000 admissions) | Duns 2014[66] | ✓ | | | | Between the Flags tool* | Australia | 1 | Y | Y | All inpatients | Uncontrolled before-after study (prospective) | 48 (24 before, 24 after) | 1058 (39.5) | 2120 (76.0) | RR=1.92 (1.79 to 2.07)† | **0.02** | 7 |
| RRT calls (per 1000 patient-days) | Panesar 2014[71] | ✓ | | ✓ | | Modified Brighton PEWS (e) | USA | 1 | Y | Y | All inpatients | Uncontrolled before-after study (retrospective) | 42 (18 before, 24 after) | 44 (3.14) | 69 (4.23) | RR=1.35 (0.92 to 1.96)† | 0.11 | 15 |
| RRT calls (per 1000 patient days) | Douglas 2016[72] | ✓ | | | ✓ | Modified Brighton PEWS (b) | USA | 1 | Y | Y | All inpatients | Uncontrolled before-after study (retrospective) | 24 (12 before, 12 after) | 194 (6.17) | 292 (9.80) | RR=1.59 (1.33 to 1.90)† | <0.001 | 12 |
| Code calls (per 1000 patient days) | Douglas 2016[72] | ✓ | | ✓ | ✓ | Modified Brighton PEWS (b) | USA | 1 | Y | Y | All inpatients | Uncontrolled before-after study (retrospective) | 24 (12 before, 12 after) | 31 (0.98) | 20 (0.67) | RR=0.68 (0.39 to 1.19)† | 0.21 | 12 |
| **PICU transfers** | | | | | | | | | | | | | | | | | | |
| Transfers from ward to other specialist units (per 1000 patient-days) | Parshuram 2011[70] | ✓ | | ✓ | ✓ | Bedside PEWS | Canada | 1 | N | N | All inpatients | Uncontrolled before-after study (prospective) | 8 (3 before, 5 after) | 5 (5.9) | 19 (8.1) | RR=1.37 (0.51 to 3.63)† | 0.54† | 23 |
| Clinical deterioration events on ward prior to transfer to specialist unit (per 1000 patient-days) | Parshuram 2011[70] | ✓ | | ✓ | ✓ | Bedside PEWS | Canada | 1 | N | N | All inpatients | Uncontrolled before-after study (prospective) | 8 (3 before, 5 after) | 2 (2.4) | 1 (0.43) | RR=0.18 (0.02 to 1.97)† | 0.16† | 23 |
| PICU transfers (per 1000 admissions) | Duns 2014[66] | ✓ | | | | Between the Flags tool* | Australia | 1 | Y | Y | All inpatients | Uncontrolled before-after study (prospective) | 48 (24 before, 24 after) | 350 (13.1) | 364 (13.1) | RR=1.00 (0.86 to 1.16)† | 0.98 | 7 |

Continued

**Table 4** Continued

| Outcome | First author, year | Intervention — Implemented a new PTTT | Implemented new RRT/MET | Modified escalation process | Staff training/education | PTTT | Country | Number of centres | Specialist unit? | Existing RRT/MET? | Population | Study design | Study duration in months | Events before, n (rate) | Events after, n (rate) | Effect size (95% CI) | P value | Quality score (max=26) |
|---|---|---|---|---|---|---|---|---|---|---|---|---|---|---|---|---|---|---|
| Unplanned PICU transfers from ward (per 1000 non-ICU patient-days) | Bonafide 2014[65] | ✓ | ✓ | | | Bedside PEWS | USA | 1 | Y | N | All inpatients | Interrupted time series study (prospective) | 59 (32 before, 27 after) | 874 (4.54) | 936 (5.25) | IRR=0.73 (0.46 to 1.14) | 0.16 | 23 |
| Unplanned transfers to PICU from ward (per 1000 patient-days) | Agulnik 2017[67] | ✓ | ✓ | | ✓ | Children's Hospital Cardiac Early Warning Score | Guatemala | 1 | Y | N | Oncology unit | Uncontrolled before-after study (retrospective) | 24 (12 before, 12 after) | 157 (9.3) | 130 (6.5) | RR=0.70 (0.56 to 0.88)† | **0.003** | 19 |
| Urgent PICU admissions (per 1000 patient-days) | Parshuram 2018[64] | ✓ | ✓ | | ✓ | Bedside PEWS | Belgium, Ireland, The Netherlands, England, Italy, Canada, New Zealand | 21 | Y | N | All inpatients | Cluster randomised trial (prospective) | 18 (6 pre, 12 post) | Con: 652 (4.01) Int: 469 (3.62) | Con: 1178 (3.83) Int: 828 (3.29) | RR=0.95 (0.82 to 1.09) | 0.45 | 23 |
| **PICU outcomes** | | | | | | | | | | | | | | | | | | |
| Critical deterioration events after PICU transfer (per 1000 non-ICU patient-days) | Bonafide 2014[65] | ✓ | ✓ | | | Bedside PEWS | USA | 1 | Y | N | All inpatients | Interrupted time series study (prospective) | 59 (32 before, 27 after) | 260† (1.35) | 282† (1.56) | IRR=0.38 (0.20 to 0.75) | **0.01** | 23 |
| Mechanical ventilation within 1 hour of unplanned PICU transfer (per unplanned transfer to PICU) | Bonafide 2014[65] | ✓ | ✓ | | | Bedside PEWS | USA | 1 | Y | N | All inpatients | Interrupted time series study (prospective) | 59 (32 before, 27 after) | 45 (5.1) | 42 (4.5) | RR=0.87 (0.58 to 1.31)† | 0.51 | 23 |
| Mechanical ventilation within 12 hours of unplanned PICU transfer (per unplanned transfer to PICU) | Bonafide 2014[65] | ✓ | ✓ | | | Bedside PEWS | USA | 1 | Y | N | All inpatients | Interrupted time series study (prospective) | 59 (32 before, 27 after) | 112 (12.8) | 103 (11.0) | IRR=0.17 (0.07 to 0.44) | **<0.001** | 23 |
| Vasopressors within 1 hour of unplanned PICU transfer (per unplanned transfer to PICU) | Bonafide 2014[65] | ✓ | ✓ | | | Bedside PEWS | USA | 1 | Y | N | All inpatients | Interrupted time series study (prospective) | 59 (32 before, 27 after) | 41 (4.7) | 16 (1.7) | RR=0.36 (0.21 to 0.64)† | **<0.001** | 23 |
| Vasopressors within 12 hours of unplanned PICU transfer (per unplanned transfer to PICU) | Bonafide 2014[65] | ✓ | ✓ | | | Bedside PEWS | USA | 1 | Y | N | All inpatients | Interrupted time series study (prospective) | 59 (32 before, 27 after) | 71 (8.1) | 57 (6.1) | IRR=0.20 (0.06 to 0.62) | **0.006** | 23 |
| Invasive ventilation in PICU (per emergency PICU admission) | Sefton 2015[68] | ✓ | | ✓ | ✓ | Modified Bristol PEWT (a) | UK | 1 | Y | N | All PICU admissions | Controlled before-after study (retrospective) | 24 (12 before, 12 after) | 118 (75.2) | 104 (62.7) | RR=0.83 (0.72 to 0.97)† | **0.002** | 16 |
| Inotropes in PICU (per emergency PICU admission) | Sefton 2015[68] | ✓ | | ✓ | ✓ | Modified Bristol PEWT (a) | UK | 1 | Y | N | All PICU admissions | Controlled before-after study (retrospective) | 24 (12 before, 12 after) | 50 (31.8) | 40 (24.1) | RR=0.76 (0.53 to 1.08)† | 0.12 | 16 |
| Intubation within 24 hours of PICU admission (per 1000 patient-days) | Agulnik 2017[67] | ✓ | ✓ | | ✓ | Children's Hospital Cardiac Early Warning Score | Guatemala | 1 | Y | N | Oncology unit | Uncontrolled before-after study (retrospective) | 24 (12 before, 12 after) | 11 (0.65) | 18 (0.90) | RR=1.38 (0.65 to 2.92)† | 0.46 | 19 |
| Vasopressors within 24 hours of PICU admission (per 1000 patient-days) | Agulnik 2017[67] | ✓ | ✓ | | ✓ | Children's Hospital Cardiac Early Warning Score | Guatemala | 1 | Y | N | Oncology unit | Uncontrolled before-after study (retrospective) | 24 (12 before, 12 after) | 29 (1.72) | 37 (1.86) | RR=1.08 (0.66 to 1.75)† | 0.60 | 19 |
| Mechanical ventilation during PICU admission (per PICU admission) | Kolovos 2018[69] | ✓ | ✓ | | | RRT activation criteria* | USA | 1 | Y | N | All unplanned PICU admissions | Uncontrolled before-after study (retrospective) | 78 (42 before, 36 after) | 285 (25.98) | 233 (22.09) | RR=0.85 (0.73 to 0.99)† | **0.03†** | 12 |
| Intubation within 1 hour of PICU admission (per PICU admission) | Kolovos 2018[69] | ✓ | ✓ | | | RRT activation criteria* | USA | 1 | Y | N | All unplanned PICU admissions | Uncontrolled before-after study (retrospective) | 78 (42 before, 36 after) | 49 (4.47) | 88 (8.34) | RR=1.87 (1.33 to 2.62) | **0.0003** | 12 |
| Significant clinical deterioration events (per 1000 patient-days) | Parshuram 2018[64] | ✓ | ✓ | | ✓ | Bedside PEWS | Belgium, Ireland, The Netherlands, England, Italy, Canada, New Zealand | 21 | Y | N | All inpatients | Cluster randomised trial (prospective) | 18 (6 pre, 12 post) | Con: 144 (0.89) Int: 80 (0.62) | Con: 259 (0.84) Int: 127 (0.50) | RR=0.77 (0.61 to 0.97) | **0.03** | 23 |

P values in bold denote statistical significance (<0.05).
A critical deterioration event is defined as transfer to the ICU followed by non-invasive or invasive mechanical ventilation or vasopressor infusion within 12 hours.[65]
*Indicates a PTTT not described or validated in the published literature.
†Data calculated by research team, based on data presented in the journal article. All data calculated via https://www.medicalc.org.
Con, control group; ED, emergency department; ICU, intensive care unit; Int, intervention group; IRR, incident risk ratio; MET, medical emergency team; N/A, not available; PICU, paediatric intensive care unit; PTTT, paediatric track and trigger tool; RR, relative risk; RRT, rapid response team.

rates of PICU mortality among ward transfers (RR=0.31, 95% CI 0.13 to 0.72),[18] and PICU mortality rates among patients readmitted within 48 hours (RR=0.43, 95% CI 0.17 to 0.99).[63] Six studies (including a high-quality cluster randomised trial and interrupted time series study) reported no postintervention change in PICU mortality using a variety of metrics.[64–69]

### Cardiac and respiratory arrests

Two uncontrolled before-after studies (both with RRT/MET) reported significant postintervention rate reductions in subcategories of cardiac arrests: one in 'near cardiopulmonary arrests'[63] (RR=0.54, 95% CI 0.52 to 0.57) but not 'actual cardiopulmonary arrests' and one in 'preventable cardiac arrests'[62] (RR=0.45, 95% CI 0.20 to 0.97) but not 'unexpected cardiac arrests'. One uncontrolled before-after study (with RRT/MET) reported a significant postintervention reduction in rates of ward respiratory arrests per 1000 patient-days[16] (RR=0.27, 95% CI 0.07 to 0.95). Seven studies (including one high-quality cluster randomised trial and one high-quality interrupted time series study) found no change in cardiac arrest rates using a variety of metrics[13 15 16 61 64 65] or cardiac and respiratory arrests combined.[60]

### Calls for urgent review/assistance

Two uncontrolled before-after studies (all with RRT/MET) reported significant postintervention reductions in rates of code calls[17 63] (RR=0.29, 95% CI 0.10 to 0.65; RR=0.71, 95% CI 0.61 to 0.83) while three studies found no change in rates of code calls.[15 18 72] One uncontrolled before-after study in a community hospital (without RRT/MET) found significant postintervention reductions in rates of urgent calls to the in-house paediatrician (RR=0.23, 95% CI 0.11 to 0.46) and respiratory therapist[70] (RR=0.36, 95% CI 0.13 to 0.95). Two uncontrolled before-after studies (with RRT/MET) found increases in rates of RRT calls[72] (RR=1.59, 95% CI 0.33 to 1.90) and outreach team calls[66] (RR=1.92, 95% CI 1.79 to 2.07). One study found no change in rates of RRT calls.[71]

### PICU transfers

One uncontrolled before-after study (without RRT/MET) found a significant postintervention decrease in the rate of unplanned PICU transfers per 1000 patient-days[67] (RR=0.70, 95% CI 0.56 to 0.88). Four studies (including one high-quality cluster randomised trial and one high-quality interrupted time series study) found no change in rates of PICU admissions postintervention.[64–66 70]

### PICU outcomes

Two studies, one interrupted time series and one multi-centre cluster randomised trial (both with RRT/MET), found significant reductions in rates of 'critical deterioration events' (life-sustaining interventions administered within 12 hours of PICU admission) relative to preimplementation trends and relative to control hospitals, respectively (IRR=0.38, 95% CI 0.20 to 0.75; OR=0.77, 95% CI 0.61 to 0.97).[64 65] One controlled before-after study (without RRT/MET) reported a significant reduction in rates of invasive ventilation given to emergency PICU admissions postintervention (RR=0.83, 95% CI 0.72 to 0.97), with no significant change observed in a control group of patients admitted to PICU from outside of the hospital.[68] One uncontrolled before-after study reported a significant postintervention decrease in rates of PICU admissions receiving mechanical ventilation (RR=0.85, 95% CI 0.73 to 0.99), but an increase in rates of early intubation (RR=1.87, 95% CI 1.33 to 2.62).[69]

### Implementation outcomes

Only three studies reported outcomes relating to the quality of implementation of the intervention. One study reported 99% of audited observation sets of the Bedside PEWS had at least five vital signs present postintervention, up from 76% preintervention (no change in control hospitals).[64] A previous study of the same PTTT reported 3% of audited cases had used the incorrect age chart but reported an intraclass coefficient of 0.90 for agreement between bedside nurses scoring the PTTT in practice and research nurses retrospectively assigned scores.[70] Finally, error rates in C-CHEWS scoring were reported to have reduced from an initial 47% to below 10% by the end of the study.[67]

## DISCUSSION

This paper reviewed the published PTTT and early warning system literature in order to assess the validity of PTTT for predicting inpatient deterioration (question 1) and the effectiveness of early warning system interventions (with or without PTTT) for reducing mortality and morbidity outcomes in hospitalised children (question 2). We believe that the consideration of broader 'early warning systems' differentiates this paper from previous reviews, as does the inclusion of two recently published high-quality effectiveness studies.[64 73]

### How well validated are existing tools for predicting inpatient deterioration?

Given a growing understanding and emphasis on the importance of local context in healthcare interventions, it is perhaps not surprising that such a wide range of PTTT have been developed and evaluated internationally, and modifications to existing PTTT are common. The result, however, is that a large number of different PTTT have been narrowly validated, but none has been broadly validated across a variety of different settings and populations. With only one exception,[44] all studies evaluating the validity of PTTT have been single-centre reports from specialist units, greatly limiting the generalisability of the findings.

PTTT such as the Bedside PEWS, C&VPEWS, NHS III PEWS and C-CHEWS have demonstrated very good (AUROC ≥0.80) or excellent (AUROC ≥0.90) diagnostic accuracy, typically for predicting PICU transfers, in internal and external validation studies.[11 14 19 29 32 37 42 44]

However, methodological issues common to the validation studies mean that such results need to be interpreted with a degree of caution. First, each of the studies was conducted in a clinical setting where paediatric inpatients are subject to various forms of routine clinical intervention throughout their admission. There are numerous statistical modelling techniques which can account for co-occurrence of clinical interventions and the longitudinal nature of the predictors,[74 75] but none of these were used in the validation studies and so estimates of predictive ability are likely to be distorted. Indeed, the majority of outcomes used in the validation studies are clinical interventions themselves (eg, PICU transfer). Second, while it understandable that a majority of studies 'bench-tested' the PTTT rather than implement it into practice before evaluation, the process of abstracting PTTT scores retrospectively from patient charts and medical records introduces a number of sources of potential bias or inaccuracy. For instance, several studies reported either high levels of missing data (ie, some of the observations required to populate the PTTT score being evaluated were not routinely collected or recorded and so were scored as 'normal')[11 19 32 44 45] or difficulty in abstracting certain descriptive or subjective PTTT components.[19 28 41 49] Assuming missing values are normal, or excluding some PTTT items for analysis are both likely to result in underscoring of the PTTT and skew the results. Finally, studies which evaluated a PTTT that had been implemented in practice are at risk of overestimating the ability of PTTT to predict proxy outcomes such as PICU transfer, inasmuch as high PTTT scores or triggers automatically direct staff towards escalation of care, or clinical actions which make escalation of care more likely.

The findings reported in several PTTT studies point towards two potential challenges for some centres in implementing and sustaining a PTTT in clinical practice. As noted above, a number of studies that retrospectively 'bench-tested' a PTTT reported that the observations that were required to score the tool were not always routinely collected or recorded in their centre. It may be that the introduction of a PTTT into practice would help create a framework to ensure that core vital signs and observations were collected more routinely (as demonstrated by Parshuram *et al*[64]), but this would obviously have resource implications that could be a potential barrier for some centres. Such considerations are important, as evidence from the adult literature points to the potential for tools to inadvertently mask deterioration when core observations are missing.[76] Second, PPV values reported in cohort studies, and case-control studies that adjusted for outcome prevalence, were uniformly low (between 2.3% and 5.9%).[14 19 31–33 47] They demonstrate that even PTTT which demonstrate good predictive performance are likely to generate a large amount of 'false alarms' because adverse outcomes are so rare. For some centres, these issues may be mitigated to some extent by dedicated response teams or other available resources, but other

hospitals may not be able to sustain the increased workload of responding to PTTT triggers.

## How effective are early warning systems for reducing mortality and morbidity?

We found limited evidence for early warning system interventions reducing mortality or arrest rates in hospitalised children. While some effectiveness papers did report significant reductions in rates of mortality (on the ward or in PICU) or cardiac arrests after implementation of different early warning system interventions,[16–18 62 63] they were all uncontrolled before-after studies which have inherent limitations in terms of establishing causality. They do not preclude the possibility that outcome rates would have improved over time regardless of the intervention[77] or changes were caused by other factors, and their inclusion is accordingly discouraged by some Cochrane review groups.[78] Three high-quality multicentre studies— two interrupted time series studies and a recent cluster randomised trial—found no changes in rates or trends of mortality or arrests postintervention.[64 65 73]

There was also limited evidence for early warning systems reducing PICU transfers or calls for urgent review. Again, a small number of uncontrolled before-after studies reported significant reductions postintervention,[15 17 63] but several other studies reported significant increases in transfers or calls for review[66 72] or no postintervention changes. We did find moderate evidence across four studies—including a controlled before-after study, a multicentre interrupted time series study and a multicentre cluster randomised trial—for early warning system interventions reducing rates of early critical interventions in children transferred to PICU.[64 65 68 69] Such results are promising, but corresponding reductions in hospital or PICU mortality rates have not yet been reported.

Implementing complex interventions in a healthcare setting is challenging and evidence from the adult literature points to challenges and barriers to successfully implement TTT in practice.[79–81] However, given so few effectiveness studies reported on implementation outcomes, it is difficult to know whether negative findings reflect poor effectiveness or implementation of early warning systems. Again, effectiveness studies were predominantly carried out in specialist centres—and in all but three cases,[67 68 70] involved the use of a dedicated response team—which greatly limits the generalisability of findings outside of these contexts.

## Limitations of the review

There are several limitations of the current review. First, despite purposely widening the scope of the effectiveness review question to include paediatric 'early warning systems' with or without a PTTT, we identified very few studies that did not employ a PTTT as part of the intervention. In part, this likely reflects the fact that PTTT have become almost synonymous with early warning systems, but it is also possible that our search strategy may have missed some broader early warning system initiatives that

were not explicitly labelled as such. Second, our inclusion criteria for study selection were deliberately broad and so resulted in our including several validation and effectiveness studies that were subsequently excluded from analysis due to insufficient statistical detail or methodological issues. Third, the scope of the current review was limited to consideration of quantitative validation and effectiveness studies. We are mindful of research suggesting that implementing PTTT in practice may confer secondary benefits including, but not limited to improvements in communication, teamwork and empowerment of junior staff to call for assistance.[82–84] Finally, we opted not to conduct a meta-analysis of effectiveness findings due to the heterogeneity of outcome metrics, interventions and study designs, populations and settings. Given the large sample sizes required to detect changes in rare adverse events, we believe further work is needed to harmonise outcome measures used to evaluate early warning system interventions internationally, in order to facilitate pooling of findings across studies.

## CONCLUSION

The PTTT literature is currently characterised by an 'absence of evidence' rather than an 'evidence of absence'. PTTT seem like a logical tool for helping staff detect and respond to deteriorating patients, but the existing evidence base is too limited to form clear judgements of their utility. We would argue that there has been too much confidence placed in the statistical findings of validation studies of PTTT, given methodological limitations in the study designs. There is evidence of consistently high false-alarm rates and bench-testing studies point to many PTTT parameters not being reliably recorded in practice: as such there is reason for caution in considering the viability of PTTT for all hospitals. Almost all of the early warning systems and PTTT reported in the literature have been developed and evaluated in specialist centres, typically in units with access to dedicated response teams—yet PTTT appear to be commonly adopted by non-specialist units with little modification. There is currently limited evidence that 'early warning systems' incorporating a PTTT reduce deterioration or death in practice. As such, we would urge caution among policymakers in calling for their use to become mandatory across all hospitals. We acknowledge the potential for PTTT to confer a range of secondary benefits in areas such as communication, teamwork and empowerment of junior staff. More work is required to understand the wider impact of PTTT implementation in different clinical settings before it is possible to evaluate their overall contribution to the wider safety mechanisms and systems aimed at identifying and responding to deteriorating in paediatric patients.

**Author affiliations**
[1]Centre for Trials Research, Cardiff University, Cardiff, UK
[2]Hull York Medical School, University of Hull, Hull, UK
[3]School of Healthcare Sciences, Cardiff University, Cardiff, UK
[4]Department of Paediatrics, Morriston Hospital, Swansea, UK
[5]Wirral University Teaching Hospital, Wirral, UK
[6]University Library Services, Cardiff University, Cardiff, UK
[7]Public Health Wales, Cardiff, UK
[8]Department of Paediatric Intensive Care, Noah's Ark Children's Hospital for Wales, Cardiff, UK
[9]SAPPHIRE Group, Health Sciences, Leicester University, Leicester, UK
[10]Paediatric Emergency Medicine Leicester Academic (PEMLA) Group, Children's Emergency Department, Leicester Royal Infirmary, Leicester, UK
[11]Alder Hey Children's NHS Foundation Trust, Liverpool, UK
[12]Faculty of Health and Applied Sciences (HAS), University of the West of England Bristol, Bristol, UK
[13]Department of Pediatric Emergency Medicine, Sidra Medical and Research Center, Doha, Qatar
[14]Division of Population Medicine, School of Medicine, Cardiff University, Cardiff, UK

**Acknowledgements** The authors would like to acknowledge the contribution of Dr James Bunn to the review.

**Contributors** RT: screening and review of papers, contribution to design of work, preparation of manuscript; CH: screening and review of papers, contribution to concept and design of work, review of manuscript; FVL-W: contribution to design of work, screening and review of papers, review of manuscript; KH: contribution to concept and design of work, screening and review of papers, review of manuscript; CP, DR, BM, AO, DE, RS, GS, DL, LNT, DA, AL, ET-J: contribution to concept and design of work, screening and review of papers, review of manuscript; MM: information specialist, review of manuscript.

**Funding** This study is funded by the National Institute for Health Research (NIHR) Health Services and Delivery Research (HS&DR) programme (12/178/17).

**Competing interests** None declared.

**Patient consent for publication** Not required.

**Provenance and peer review** Not commissioned; externally peer reviewed.

**Data sharing statement** All data relevant to the study are included in the article or uploaded as supplementary information.

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
