## [Reviewer comments · BMJ Open]

ARTICLE DETAILS

TITLE (PROVISIONAL)	Validity and effectiveness of paediatric early warning systems and track and trigger tools for identifying and reducing clinical deterioration in hospitalised children: a systematic review
AUTHORS	Trubey, Rob; Huang, Chao; Lugg-Widger, Fiona; Hood, Kerenza; Allen, Davina; Edwards, Dawn; Lacy, David; Lloyd, Amy; Mann, Mala; Mason, Brendan; Oliver, Alison; Roland, Damian; Sefton, G.; Skone, Richard; Thomas-Jones, Emma; Tume, Lyvonne N.; Powell, Colin

VERSION 1 - REVIEW

REVIEWER	Dr Veronica Lambert Dublin City University Ireland
REVIEW RETURNED	16-Feb-2018

GENERAL COMMENTS	Thank you for inviting me to review this manuscript which reviewed the evidence base for both the validation and impact of PTTT. The review is comprehensive and clearly presented and offers a valuable contribution to the existing evidence base on PEWS for identifying deterioration of children in hospital. While the review finding largely support two other recent reviews in this field this manuscripts offers another perspective and support the drive to consider the wider system into which PTTT are embedded. The only minor comment I would have is that the authors do not report on any limitations to the conduct of their own review.
---

REVIEWER	Anne Lee Solevåg Akershus University Hospital, Norway
REVIEW RETURNED	17-Feb-2018

GENERAL COMMENTS	Dear Editor of BMJ Open and the authors of original paper entitled 'Systematic review of paediatric track and trigger tools for identifying clinical deterioration of children in hospital: their development, validation and effectiveness' by Trubey and coworkers. Thank you for the opportunity to review this systematic review of 35 validation studies and 24 effectiveness studies of paediatric track and trigger tools. The authors conclude that due to the lack of high quality studies demonstrating benefit, recommendations for universal use cannot be justified. More research is needed.
--

The authors have clearly put down a large amount of work performing this review and the methods seem sound and of good quality. However, I disagree with some interpretations of the included articles, and with the overall conclusion of the paper. In addition, I am concerned that the authors did not perform a more recent updated literature search, as there is a continuously growing body of evidence about paediatric early warning tools. (The authors should mention that the main search was carried out in Jan 2015, not only in the appendix, but in the manuscript's methods section)

Below are some comments to the different sections of the manuscript:

Under the heading STRENGTHS AND LIMITATIONS OF THIS STUDY, I would expect to find strengths and weaknesses of the review itself, not the included studies.

In general, I think that the authors did not address potential limitations of their review in the manuscript

Abstract

Objective:

'To assess (1) how well validated existing paediatric track and trigger tools (PTTT) are, and (2) how effective paediatric early warning systems (with or without a PTTT) are at reducing mortality and morbidity outcomes in hospitalised children.'

-What is meant by 'paediatric early warning systems (with or without a PTTT)'? It would be useful if the authors could explain what the difference is between paediatric early warning systems and PTTT? (Chapman et al explain the difference in their review). This applies to the main body of the manuscript as well. The distinction between paediatric early warning systems and PTTT is particularly unclear as it is stated in the introduction that the 'umbrella term 'paediatric track and trigger tools' (PTTT)' is used. Thus, 'paediatric early warning systems (with or without a PTTT)' is a confusing sentence

Methods

The greatest limitation of this study is the fact that the search was completed September 2016 (Chapman et al, reference number 8 conducted their search in May 2016; Lambert et al, reference number 9, in August 2016), which is 17 months ago. Although the authors included unpublished and in press studies, I expect that significant studies may have been missed in this review. Through a quick Pubmed search (Febr 15 2018) using the term 'paediatric early warning score', a number of more recent hits of potential significance were retrieved.

References number 8 and 9 are quite recent systematic reviews of pediatric early warning scores. What does this particular systematic review add? Please explain

The authors state in the first paragraph of the discussion section that: 'We believe these specific questions have not been answered in previous reviews'. Can they be more specific?

Methods page 7:

I don't understand the meaning of the sentence: 'A data extraction form was developed for the two questions which had some common elements relating to both research questions'

Results page 10

The authors refer to the Bristol PEWS and the Bristol PEWT. Is this an error? Please also include reference numbers for clarity

Results page 13:

I suggest that the following sentence be removed as it does not provide evidence of incompleteness of PEWS scoring, but incompleteness of an extended observation form (without scoring):

'Edwards et al.³⁶ did not implement their PTTT, but modified observation charts to include all eight items in the Cardiff & Vale PEWS. They reported an average completion rate of 44% for the different parameters.'

-Such a use and interpretation of results may have lead the authors to (falsely) conclude in the discussion section page 17 that: 'Each PTTT pre-supposes that several vital signs and observations are regularly completed in order to properly score patients. However, this appears to be uncommon, and evidence from the adult literature points to the potential for tools to inadvertently mask deterioration when core observations are missing'

Discussion page 18:

The sentence: 'This occurred alongside a parallel increase in overall PICU admissions, and so it is not clear whether the PTTT specifically improved identification of at-risk children, or rather encouraged a broadly more conservative approach to admitting children to the ICU'

-Should it be a more liberal (not conservative) approach?

Conclusion

I think the authors should take into consideration benefits of using scoring tools, e.g. a) documenting vital signs and b) a more clear and precise communication between health care personnel. When they conclude that PTTT cannot be universally recommended, I disagree because of the significant secondary benefits mentioned. Lambert et al address these issues in their review.

Tables

Should the abbreviations used in the tables be explained in table legends?

I think that it would be useful to include reference numbers (not just the last name of the first author) in the tables

In conclusion, the authors have made a substantial effort summarizing the current evidence about PEWS/PTTT. However, their interpretations of the evidence and their conclusion could be

	more nuanced. Importantly, they should be significantly clearer about what this paper adds to other recent systematic reviews on the topic.
--	---

REVIEWER	Dr Sue Chapman Great Ormond Street Hospital UK
REVIEW RETURNED	08-Mar-2018

GENERAL COMMENTS	Thank you for asking me to review this manuscript. It is on a area which will be of interest to the readers of BMJ open who practice in paediatrics. The overall quality of the writing in terms of flow is good and the discussion is balanced. However I do have some concerns about the methodology and results which I will discuss below. Abstract: Appropriate and summarises the paper. Background: Appropriate summary of the literature. Methods: I do have some concerns over the methodology and results reporting. PRISMA is guideline on reporting systematic reviews rather than the method of conducting the review itself. Many journals and Cochrane now recommend the GRADE methodology as the most rigorous approach. This incorporates the PICOS criteria, which would also be helpful in making the research question more transparent. The research questions themselves are a little vague - For what population? Where? Fundamentally what does 'how well validated' actually mean? In terms of number of studies, quality of studies, number of subjects? Or are you looking at the actual ability of the PTTT to accurately detect children at risk? The research question doesn't quite make sense to me I'm afraid. GRADE also looks at quality of evidence across the body of research (rather than each individual paper) with a structured and transparent assessment process and presents summary of findings across the body of research as a whole, which is generally more useful for clinicians. The methods section reports that modified version of the Down and Blacks was used to assess quality. I could not see this presented anywhere in the paper so it is difficult to know how rigorous this is. It would also be helpful to see the data extraction forms as supplementary data. There is no indication what the reviewers screened the papers for and no inclusion and exclusion criteria. Results: Figure 1 is labelled as the 'PRISMA flow diagram' however it is probably more accurate to say it is the flow diagram of the results of the search. The strategies for identifying 'other sources' needs explanation. Table 1 is helpful and well presented. Question 1: It would be of great interest to know how exactly many different PTTS were found overall, rather than 'a range'. It would be helpful to know the numbers of PTTT that were based on each of the four PTTS (rather than 'the vast majority'). How was it determined that these were developed from the 4 'main' PTTT? Was there an assessment process? How was this conducted? It would also be helpful to know how many PTTS were age-dependent and age-independent.
---

Table 2: Three items in the table do not appear in any of the PTTS so I am not sure of their significance. I am also not sure of the value of this table, given that it only describes 4 of the PTTS identified in the 59 papers. Without summarising the whole data the reader cannot tell how prevalent each vital sign or observation was across the dataset as a whole.

Page 12, lines 7 onwards - AUROC, sensitivity...:'Few exceptions' is too vague, particularly as then only one reference is cited. As the research question states that the component parts will be examined, it needs a bit more description.

Table 3: Needs some description of how the papers are ordered. They appear to be grouped together under each of the 'main PTTS'. Without knowing how similar or dis-similar they are to the 'original' PTTS its hard to know if this makes sense. Grouping by outcome makes more clinical sense. Again this may be related to the vagueness of the research question. There is no quality assessment of the papers presented and no indication of how large the studies were. The footnote says that many papers could not be included as they did not present sufficient statistical detail - how did they meet the eligibility criteria to be a part of the validation studies group if no data was presented?

The table also does not describe the acronyms used (PICU, PHDU) etc and what is meant by threshold tested. No confidence intervals are presented, which would again help with assessing significance.

Inter-rater reliability: Again, it is quite vague to say 'some PTTS contain subjective judgements'. How many? which ones? The term 'missing data' implies missing data within the context of the research studies, however this appears to relate more to incomplete scoring of the PTTT themselves, which is a different issue. There are also some studies which assessed reliability of PTTS scoring which do not appear in the list. Reliability is distinct from validity so I am not sure how this section relates to your research question. They are reported for a number of papers which don't appear in your list (Parsharam 2011 and Randhawa and Demmel) but it is not clear why.

Page 13 line 27 onwards - 'typically multi-faceted' - typical in what way? Multi faceted in what way?

'Several unpublished activation criteria - (references 44-50) how did you classify these as unpublished as they have references cited?

There is no link to Supplementary Table 2 which presents the study characteristics of the validation studies. This would appear to be important data that would help evaluate the SR findings. I'm not sure this should be supplemental data only.

There are a lot of gaps in the table, particularly around study design, primary outcome measure, sample sizes. Its not clear if the data doesn't exist or is not reported. There is no quality assessment reported. There is also a section marked 'full-text'. Your flow chart (figure 1) implies that the full-text of all the included studies was obtained, but Supp Table 2 implies differently. This is an important methodological issue.

There is also no link to the reference for each paper on Supp Table 2 or 3, which would be useful. Parsharam published 2 papers in 2011 so these are needed. There is also no indication of how the studies were ordered. Is it significant that Duncan is first?

Supplementary Table 5 has similar issues with some papers having no reported outcomes, however being classified as 'effectiveness' studies? How was this done if there are no outcomes? Again, no quality assessment.

Page 30 - Table 4 has similar problems with no quality assessment and no indication of the size of the study. The outcomes are just reported as arrows, with no values given, so it does not help the reader to know the significance of the findings. There is no definition of critical interventions. Where you say in the footnote that a number of papers (how many?) did not report significant reductions - did they report non-significant findings? Or increases? Can these results be pooled? If not, why not?? Some papers have unpublished activation criteria - how they came to be included in the study needs more explanation.

Your research questions implies you will examine PEWS (with and without a PTTS) but you present these results all together in one Table (and I also wonder if you mean PTTS (with and without a PEWS), not the other way round. Either way, PTTS and PEWS need definition.

Page 16 - Discussion:

Line 22-44 I am not sure that we know that in a hypothetically perfect hospital there would be no statistical relationship between a high PTTT score and adverse outcome. If PTTT are predictive then there may be clinical signs that cannot be corrected by immediate clinical intervention and these will always continue to display abnormality and lead to death or arrest. We simply don't know as its not been studied. So I agree with your caution in how we interpret these results, but not in how you've explained it.

The other limitation are valid and well described.

Your discussion is more on the actually validity of PTTT rather than how well validated they are (which is more an expression of the quality of the research to me)

Page 17 line 44 onwards -How effective?: You note the difficulty of studies which also implement a response team and/or training. It would be helpful to know why you then considered these studies 'all together' rather than examining them separately.

Within effectiveness you also discuss fidelity or adherence. I agree with the challenges, limitations and lack of research however you need to explain more in either your introduction what exactly you mean by effectiveness.

Conclusion: Appropriate and agree that PTTT should be used with caution.

Overall my main criticisms are the lack of rigour in the review, both methodologically and in the reporting. There is a lot of subjective description rather than objective findings (for example 'the vast majority were based on 4 PTTT, with few exceptions etc). Table 4 has arrows to indicate the effect on outcomes, rather than reporting what was found. It doesn't give the reader confidence that you know exactly how many PTTS were based on the 4 PTTT or what the research findings were and this undermines the rigour of your assessment

There is a lack of transparency in the methodology overall. PRISMA is designed to assist with the reporting not with methodology. I would recommend you consider the GRADE methodology as it is rigorous and transparent. You do not report anything on the quality of either the individual papers or (more

	usefully) the quality of the body of evidence as a whole. Papers appear to be included where the full text was not obtained. This raises issues around the criteria you applied when selecting the papers (which is not reported). The research questions also do not make complete sense to me - (what is well validated?) and do not conform to PICOS standards. The Tables are hard to follow and incomplete with missing data. There is no reporting of either the quality method you used or the findings. I can see a lot of work has gone into identifying the papers and thinking about how to approach this. The search strategy is comprehensive (although the search is from 18/12 ago). The majority of the writing is good in terms of flow. However it doesn't feel rigorous and there is no attempt to synthesis and analyse the data as a whole. Sorry to be so harsh....
--	--

REVIEWER	Jacqueline Birks University of Oxford UK
REVIEW RETURNED	26-Apr-2018

GENERAL COMMENTS	The objectives of this systematic review are to assess the validation of existing Paediatric Track and trigger Tools (PTTT), and secondly to assess the results of testing PTTTs, whether they are effective at reducing mortality and morbidity outcomes in children in hospital. Background There is the briefest description of a PTTT. As the PTTT is central to the review there should be a description of the physiological measures that can be included, plus other measurements. Methods of collection of the measurements, by paper charts, or electronically, are not described. The scores are usually a simple sum of weights accorded to the individual measurements, but this is not described. Because the review is about validation we could expect some description of the tools in use and how they were derived, that is by a statistical model, or by a committee of experts. If they are based on prediction algorithms, how large was the derivation data set? There appears to be a small number of PTTTs, but these are often modified in use. A modified PTTT should be considered a separate PTTT and the method of derivation described. If there is no information on the derivation of the modified PTTT, or if it appears to have been introduced by the simple addition of additional predictors or covariates without any evidence then this should be described. We should be able to judge the quality of the PTTTs in use and this is best reported in a table. Two recent systematic reviews of published PTTT are cited, but it is not clear why this new review is needed as there is no critical assessment of the existing reviews. Chapman (2016) appears to be a high quality review with a much better background to PTTT than this review. Lambert (2017) was also conducted to a high standard. Table 3 Summary of PTTT validation study findings It is very difficult validating an early warning score. From the information reported in the tables we know the number of patients in the validation study, the PTTT being validated, the study design, outcome measure and results from the measure of discrimination.
--

	There is much additional information that would allow us to judge each validation study. We do not know whether the studies are internal validation studies or external. Each patient in a study may have several assessments of the PTTT, how was this dealt with in the validation? There is no quality assessment of the studies, although the PRISMA check list indicates that this was done and reported, and the methods section indicates that it will be carried out (Downs and Black rating scale). Downs and Black (1998) report a feasibility study for a quality assessment rating scale for study quality, it is not a report of a tried and tested quality rating scale. The authors should assess the quality of the included studies. I have looked at two included studies. Duncan (2006) appears to be a study deriving a PTTT, not validating an existing PTTT. Mandell (2015) reports a validation study of the Brighton EWS, but a version modified at the hospital where the study validation is carried out. Further information is required about the derivation of the modified version. It would be helpful to report whether each included study reported valid statistics of discrimination and calibration relating to the PTTT that was being validated. Table 4 does not reported results, although according to the footnote studies whose results were not statistically significant were excluded from the table, so these results are available. Again a quality rating of these studies is important. The quality rating should include an assessment of the quality of the PTTT being tested In conclusion the authors have completed a complex review of this difficult topic. There appear to be a small number of early warning scores. Many of the included studies appear to be validating or testing modifications to the original EWS. There is no information in the review on the derivation of modifications. The review would be improved by presenting more information on the modifications to scores. It is important to measure the quality of the included studies. The authors should derive a list of factors to assess the quality and a scoring system to derive an overall quality score. This would provide a better overview of this very heterogeneous collection of studies.
--	--

VERSION 1 – AUTHOR RESPONSE

Reviewer 1

The only minor comment I would have is that the authors do not report on any limitations to the conduct of their own review.

We have amended the ‘Strengths & Weaknesses’ section to include some study weaknesses and added a sub-section to the Discussion on study limitations.

Reviewer 2

I am concerned that the authors did not perform a more recent updated literature search, as there is a continuously growing body of evidence about paediatric early warning tools. (The authors should mention that the main search was carried out in Jan 2015, not only in the appendix, but in the manuscript’s methods section)

We have updated the search through May 2018, and explained this more clearly in the Methods section.

Under the heading STRENGTHS AND LIMITATIONS OF THIS STUDY, I would expect to find strengths and weaknesses of the review itself, not the included studies.

In general, I think that the authors did not address potential limitations of their review in the manuscript

As above, we have amended the 'Strengths & Weaknesses' section accordingly and added a section on limitations to our study in the Discussion section.

What is meant by 'paediatric early warning systems (with or without a PTTT)'? It would be useful if the authors could explain what the difference is between paediatric early warning systems and PTTT? (Chapman et al explain the difference in their review). This applies to the main body of the manuscript as well. The distinction between paediatric early warning systems and PTTT is particularly unclear as it is stated in the introduction that the 'umbrella term 'paediatric track and trigger tools' (PTTT)' is used. Thus, 'paediatric early warning systems (with or without a PTTT)' is a confusing sentence

We have re-written a number of paragraphs in the introduction to be clearer on the way in which we've used these terms: we refer to a broader 'paediatric early warning system' as being "any patient safety initiative or programme used in a paediatric inpatient unit which aims to monitor, detect and respond to signs of deterioration quickly in order to avert adverse outcomes and premature death" and a 'paediatric track & trigger tool' as being one specific form of early warning system initiative.¹

We recognise that definition of terms is particularly important in this area, where terminology and acronyms describing tools, scores and systems has often been used interchangeably..

The greatest limitation of this study is the fact that the search was completed September 2016 (Chapman et al, reference number 8 conducted their search in May 2016; Lambert et al, reference number 9, in August 2016), which is 17 months ago. Although the authors included unpublished and in press studies, I expect that significant studies may have been missed in this review. Through a quick Pubmed search (Febr 15 2018) using the term 'paediatric early warning score', a number of more recent hits of potential significance were retrieved.

As above, the search is now updated through May 2018.

References number 8 and 9 are quite recent systematic reviews of pediatric early warning scores. What does this particular systematic review add? Please explain

The authors state in the first paragraph of the discussion section that: 'We believe these specific questions have not been answered in previous reviews'. Can they be more specific?

As above, we have been more explicit in explaining what we think this review offers above the existing literature. This of course now includes the recency of the search, given a number of important recent publications in this area.

I don't understand the meaning of the sentence: 'A data extraction form was developed for the two questions which had some common elements relating to both research questions'

This section has been re-written for clarity.

The authors refer to the Bristol PEWS and the Bristol PEWT. Is this an error? Please also include reference numbers for clarity

This was a typo and has been corrected. Reference numbers to studies are now provided in all results tables.

I suggest that the following sentence be removed as it does not provide evidence of incompleteness of PEWS scoring, but incompleteness of an extended observation form (without scoring):

'Edwards et al.³⁶ did not implement their PTTT, but modified observation charts to include all eight items in the Cardiff & Vale PEWS. They reported an average completion rate of 44% for the different parameters.'

-Such a use and interpretation of results may have lead the authors to (falsely) conclude in the discussion section page 17 that: 'Each PTTT pre-supposes that several vital signs and observations are regularly completed in order to properly score patients. However, this appears to be uncommon, and evidence from the adult literature points to the potential for tools to inadvertently mask deterioration when core observations are missing'

This particular sentence has been removed, and we agree that it is important to distinguish between studies which implemented a score in practice and those who abstracted data from charts and medical records ('bench testing'). However, we still believe that missing data (both in studies where scores have been used in practice, and those where they are bench tested) are an important methodological consideration for accurately evaluating PTTT performance – and also, that studies showing that core vital signs are often not always reliably documented point towards a potential barrier to successfully implementing PTTT in practice.

The sentence: 'This occurred alongside a parallel increase in overall PICU admissions, and so it is not clear whether the PTTT specifically improved identification of at-risk children, or rather encouraged a broadly more conservative approach to admitting children to the ICU'

-Should it be a more liberal (not conservative) approach?

We have re-written this part of the Discussion, and the above sentence is now removed.

I think the authors should take into consideration benefits of using scoring tools, e.g. a) documenting vital signs and b) a more clear and precise communication between health care personnel. When they conclude that PTTT cannot be universally recommended, I disagree because of the significant secondary benefits mentioned. Lambert et al address these issues in their review.

We acknowledge this point and have reflected in the limitations section of the Discussion that inclusion of studies considering wider benefits (including qualitative studies) were beyond the scope of our research questions. However, we have referenced these potential secondary benefits and hope that this gives a more nuanced conclusion. Our team have been working on a parallel review which takes a wider view of the potential impact of PTTT on the wider clinical microsystem which is close to submission.

Should the abbreviations used in the tables be explained in table legends?

I think that it would be useful to include reference numbers (not just the last name of the first author) in the tables

We have added explanation of all abbreviations used in the tables, and added reference numbers to each table as well.

Reviewer 3

I do have some concerns over the methodology and results reporting. PRISMA is guideline on reporting systematic reviews rather than the method of conducting the review itself. Many journals and Cochrane now recommend the GRADE methodology as the most rigorous approach. This incorporates the PICOS criteria, which would also be helpful in making the research question more transparent. The research questions themselves are a little vague - For what population? Where? Fundamentally what does 'how well validated' actually mean? In terms of number of studies, quality of studies, number of subjects? Or are you looking at the actual ability of the PTTT to accurately detect children at risk? The research question doesn't quite make sense to me I'm afraid.

We have incorporated the PICOS criteria into the methods section of the paper – further defining the inclusion criteria for the validation and effectiveness papers and hopefully adding more clarity to the research question in terms of what population and where. We have amended the validation research question to: “How well validated are existing paediatric track and trigger tools (PTTT) and their component parts for predicting in-patient deterioration?” and added this clarification elsewhere.

GRADE also looks at quality of evidence across the body of research (rather than each individual paper) with a structured and transparent assessment process and presents summary of findings across the body of research as a whole, which is generally more useful for clinicians.

We recognise the value of the GRADE methodology, but we are also mindful of maintaining consistency with what we originally set out to do and documented in our Systematic Review protocol published on Prospero. We have however added the details of our quality assessment scores of each study, and synthesise findings from the wider body of research in the results and discussion section, with reference to study quality and risk of bias.

The methods section reports that modified version of the Down and Blacks was used to assess quality. I could not see this presented anywhere in the paper so it is difficult to know how rigorous this is. It would also be helpful to see the data extraction forms as supplementary data.

There is no indication what the reviewers screened the papers for and no inclusion and exclusion criteria.

We have presented the quality assessment scores in the results tables, and have added the quality assessment scales to the supplementary materials. The PICOS criteria are also referred to in the methods section (and presented in full in supplementary tables), hopefully clarifying the inclusion and exclusion criteria used for screening.

Question 1: It would be of great interest to know how exactly many different PTTS were found overall, rather than 'a range'. It would be helpful to know the numbers of PTTT that were based on each of the four PTTS (rather than 'the vast majority'). How was it determined that these were developed from the 4 'main' PTTT? Was there an assessment process? How was this conducted?

It would also be helpful to know how many PTTS were age-dependent and age-independent.

We have added a table detailing each of the unique PTTT (Table 2), and been specific about how many we identified in the text of the results. This table clarifies which PTTT are based on the Brighton PEWS, Melbourne criteria, Bristol PEWT and Bedside PEWS and again this is now clarified in the text. There is a column in this table detailing which PTTT are age-dependent and age-independent and this is also referred to in the text.

Table 2: Three items in the table do not appear in any of the PTTS so I am not sure of their significance. I am also not sure of the value of this table, given that it only describes 4 of the PTTS

identified in the 59 papers. Without summarising the whole data the reader cannot tell how prevalent each vital sign or observation was across the dataset as a whole.

This table has been removed and replaced with the far more detailed table summarising PTTT mentioned above (Table 2).

Table 3: Needs some description of how the papers are ordered. They appear to be grouped together under each of the 'main PTTS'. Without knowing how similar or dis-similar they are to the 'original' PTTS its hard to know if this makes sense. Grouping by outcome makes more clinical sense. Again this may be related to the vagueness of the research question. There is no quality assessment of the papers presented and no indication of how large the studies were. The footnote says that many papers could not be included as they did not present sufficient statistical detail - how did they meet the eligibility criteria to be a part of the validation studies group if no data was presented? The table also does not describe the acronyms used (PICU, PHDU) etc and what is meant by threshold tested. No confidence intervals are presented, which would again help with assessing significance.

This table (Table 3) has been completely updated, and now includes details on quality assessment scores. An explanation of why certain papers met inclusion criteria but were excluded from analysis is given in the text (and details of excluded papers, including full reason for exclusion, are provided in a supplementary table). Abbreviations are listed at the footer of the table- and we've changed the wording of the 'threshold tested' column heading for clarity.

Inter-rater reliability: Again, it is quite vague to say 'some PTTS contain subjective judgements'. How many? which ones? The term 'missing data' implies missing data within the context of the research studies, however this appears to relate more to incomplete scoring of the PTTT themselves, which is a different issue. There are also some studies which assessed reliability of PTTS scoring which do not appear in the list. Reliability is distinct from validity so I am not sure how this section relates to your research question. They are reported for a number of papers which don't appear in your list (Parsharam 2011 and Randhawa and Demmel) but it is not clear why.

We are now more explicit about exactly which papers refer to reliability of scoring (including the papers listed) – and have amended reference to 'missing data' to 'completeness of scoring'. We would argue that reliability and completeness of data are important aspects of judging the quality of the findings of validation studies – and so think it is important to draw attention to these in the table and text.

Page 13 line 27 onwards - 'typically multi-faceted' - typical in what way? Multi faceted in what way?

'Several unpublished activation criteria - (references 44-50) how did you classify these as unpublished as they have references cited?

There is no link to Supplementary Table 2 which presents the study characteristics of the validation studies. This would appear to be important data that would help evaluate the SR findings. I'm not sure this should be supplemental data only.

There are a lot of gaps in the table, particularly around study design, primary outcome measure, sample sizes. Its not clear if the data doesn't exist or is not reported. There is no quality assessment reported.

We have amended this section to be clearer about the precise nature of each of the interventions, and to be much clearer about which PTTT used in the effectiveness studies were and were not validated. We have overhauled Table 3 to include more details of study characteristics – there are no gaps in the table. Quality assessment scores are reported for all studies.

There is also a section marked 'full-text'. Your flow chart (figure 1) implies that the full-text of all the included studies was obtained, but Supp Table 2 implies differently. This is an important methodological issue.

The 'full text' column referred to whether the study in question was a conference abstract or full text article – we acknowledge that this wasn't clear and have removed this column. We have removed Supp Table 2 and put more study characteristic detail in to the main effectiveness results table (Table 4).

There is also no link to the reference for each paper on Supp Table 2 or 3, which would be useful. Parsharam published 2 papers in 2011 so these are needed. There is also no indication of how the studies were ordered. Is it significant that Duncan is first?

Supplementary Table 5 has similar issues with some papers having no reported outcomes, however being classified as 'effectiveness' studies? How was this done if there are no outcomes? Again, no quality assessment.

See comments above.

Page 30 - Table 4 has similar problems with no quality assessment and no indication of the size of the study. The outcomes are just reported as arrows, with no values given, so it does not help the reader to know the significance of the findings. There is no definition of critical interventions. Where you say in the footnote that a number of papers (how many?) did not report significant reductions - did they report non-significant findings? Or increases? Can these results be pooled? If not, why not?? Some papers have unpublished activation criteria - how they came to be included in the study needs more explanation.

The issues raised above have been addressed with an overhaul of Table 4. All outcomes reported for the included effectiveness papers are now listed in the table (with effect size and P values), which is arranged by outcome measure.

We decided against pooling results by outcome, due to the heterogeneity of the studies – in terms of exact metrics and denominators used, settings, study populations, study designs, PTTT used, etc.

Your research questions implies you will examine PEWS (with and without a PTTS) but you present these results all together in one Table (and I also wonder if you mean PTTS (with and without a PEWS), not the other way round. Either way, PTTS and PEWS need definition.

We have clarified the use of these terms in the introduction and in the opening paragraphs of the effectiveness results (see above). The effectiveness results table (Table 4) clarifies the exact nature of each study intervention.

Page 16 - Discussion:

Line 22-44 I am not sure that we know that in a hypothetically perfect hospital there would be no statistical relationship between a high PTTT score and adverse outcome. If PTTT are predictive then there may be clinical signs that cannot be corrected by immediate clinical intervention and these will always continue to display abnormality and lead to death or arrest. We simply don't know as its not been studied. So I agree with your caution in how we interpret these results, but not in how you've explained it.

We have amended this paragraph to be clearer about the point we were trying to make – that we should be cautious in interpreting the results of validation studies (in terms of AUROC, sensitivity, etc) because of the challenges of doing this sort of research in real-world clinical settings where clinical interventions are ongoing but not controlled for.

Page 17 line 44 onwards -How effective?: You note the difficulty of studies which also implement a response team and/or training. It would be helpful to know why you then considered these studies 'all together' rather than examining them separately.

Within effectiveness you also discuss fidelity or adherence. I agree with the challenges, limitations and lack of research however you need to explain more in either your introduction what exactly you mean by effectiveness.

We acknowledge that studies which implement a response team/training package alongside a PTTT can be considered qualitatively different from those that implement a PTTT without either. However, the latter were comparatively rare (we only included 3 of these studies in our effectiveness results) so we decided to include them in the same table but make very clear which studies implemented what. We do highlight the lack of evidence for PTTT in isolation in the discussion – in terms of the (lack of) generalisability of many of the effectiveness findings to contexts such as the UK's district general hospitals where no dedicated response teams exist.

Reviewer 4

There is the briefest description of a PTTT. As the PTTT is central to the review there should be a description of the physiological measures that can be included, plus other measurements. Methods of collection of the measurements, by paper charts, or electronically, are not described. The scores are usually a simple sum of weights accorded to the individual measurements, but this is not described.

We have added some more detail to the description in the introduction (while being mindful of word count). However, the addition of Table 2 – describing in detail the components of each of the PTTT identified in the literature – hopefully addresses the reviewer's substantive point.

Because the review is about validation we could expect some description of the tools in use and how they were derived, that is by a statistical model, or by a committee of experts. If they are based on prediction algorithms, how large was the derivation data set? There appears to be a small number of PTTTs, but these are often modified in use. A modified PTTT should be considered a separate PTTT and the method of derivation described. If there is no information on the derivation of the modified PTTT, or if it appears to have been introduced by the simple addition of additional predictors or covariates without any evidence then this should be described. We should be able to judge the quality of the PTTTs in use and this is best reported in a table.

Table 2 addresses the above points. It gives details about all of the unique PTTTs identified, including the underlying components and the way in which each PTTT was developed or modified.

Two recent systematic reviews of published PTTT are cited, but it is not clear why this new review is needed as there is no critical assessment of the existing reviews. Chapman (2016) appears to be a high quality review with a much better background to PTTT than this review. Lambert (2017) was also conducted to a high standard.

As above, we acknowledge the recent SRs but feel that the current review differs enough – and presents enough new information – as to add to the literature. We have addressed this point briefly in the discussion.

Table 3 Summary of PTTT validation study findings

It is very difficult validating an early warning score. From the information reported in the tables we know the number of patients in the validation study, the PTTT being validated, the study design, outcome measure and results from the measure of discrimination. There is much additional

information that would allow us to judge each validation study. We do not know whether the studies are internal validation studies or external. Each patient in a study may have several assessments of the PTTT, how was this dealt with in the validation?

There is now a column in Table 3 which clarifies which studies were internal or external validation studies – and also a column which clarifies how studies dealt with patients having multiple assessments of the PTTT over the course of their admission.

There is no quality assessment of the studies, although the PRISMA check list indicates that this was done and reported, and the methods section indicates that it will be carried out (Downs and Black rating scale). Downs and Black (1998) report a feasibility study for a quality assessment rating scale for study quality, it is not a report of a tried and tested quality rating scale. The authors should assess the quality of the included studies.

As above, quality assessment scores are now included, and we have added a template of the quality assessment forms for validation and effectiveness studies to the supplementary materials.

I have looked at two included studies. Duncan (2006) appears to be a study deriving a PTTT, not validating an existing PTTT. Mandell (2015) reports a validation study of the Brighton EWS, but a version modified at the hospital where the study validation is carried out. Further information is required about the derivation of the modified version. It would be helpful to report whether each included study reported valid statistics of discrimination and calibration relating to the PTTT that was being validated.

As above, Table 2 now gives details of how each PTTT was derived or modified – and Table 3 clarifies which studies are internal or external validation studies.

Table 4 does not reported results, although according to the footnote studies whose results were not statistically significant were excluded from the table, so these results are available.

Again a quality rating of these studies is important. The quality rating should include an assessment of the quality of the PTTT being tested

As above, Table 4 has been overhauled to include details (including effect sizes, p values, etc) for all outcomes reported across the effectiveness studies, whether significant or not. Again, we have added quality assessment scores to the table for each study.

The review would be improved by presenting more information on the modifications to scores. It is important to measure the quality of the included studies. The authors should derive a list of factors to assess the quality and a scoring system to derive an overall quality score. This would provide a better overview of this very heterogeneous collection of studies.

Table 2 now lists details of modifications to PTTT. And quality scores have been added for both validation and effectiveness studies.

VERSION 2 – REVIEW

REVIEWER	Anne Lee Solevåg The Department of Paediatric and Adolescent Medicine, Akershus University Hospital, Norway
REVIEW RETURNED	08-Oct-2018

GENERAL COMMENTS	Dear Editor of BMJ Open and the authors of original paper entitled 'Validity and effectiveness of paediatric early warning systems and track and trigger tools for identifying and reducing clinical deterioration in hospitalised children: a systematic review' by Trubey and coworkers. I absolutely love the revised version of the review. The results as they are now presented are important contributions that should guide future research and practice. Thank you and congratulations.
--

REVIEWER	Dr. Sue Chapman Great Ormond Street Hospital Great Ormond Street London UK
REVIEW RETURNED	24-Oct-2018

GENERAL COMMENTS	Thank you for asking me to review this paper. It represents a huge amount of work on the part of the authors to summarise and synthesis the literature on PTTT and PEWS. In the main it is well written and comprehensive. I do have some feedback which I hope the authors will find helpful. Abstract: Appropriate summary of the study Strengths and limitations: Appropriate Background: Good summary and introduction Methods: Overall describes the study clearly. Page 7, Line 6-8: I would question whether PRISMA is a methodology to be followed when conducting a SR. It is a minimum set of items for reporting a SR, but does not give guidance on conducting one. GRADE is acknowledged by many journals as the most robust method of conducting a SR. GRADE also allows recommendations to made regarding the strength of the literature as a whole rather than just the individual component studies. It would help the reader if you could justify why you chose this method rather than GRADE or other more recognised SR methodologies. P7, L 53: acuity at PICU admission and PICU outcomes stated here but not on PICOS (Supplementary table 2) P8 L8-10 - may be helpful to see data extraction forms as supplemental data if space allows P8 L21: Mainly reported 95% CI not p values. Also reported OR and other analyses which need including here. Your description of data analysis feels a little sparse. Quality appraisal: This is probably my main criticism of the conduct of the study. Downs and Black is now 20 years old and it is not clear why you chose this above other quality measures. It is also not clear how this was modified (the original paper reports 27 items but your modified tool contains 12. This links with my feedback above regarding the methodology chosen. Assessing the quality of individual papers, whilst helpful, does not make it clear to the reader the strength of the evidence 'as a whole' and limits the ability to truly synthesise the literature. PPI: A welcome addition to a SR.
--

Results: Largely well presented.

Table 1: The section on study design does not present items in rank order which is inconsistent with the rest of the table.

Table 2: This is probably my other area of concern. Your tables are really long and quite labour intensive on the part of the reader. If it is at all possible to reduce down the wording, particularly in the 3rd column, it would make it more digestible. The information is helpful, but you just get a bit overloaded.

In terms of headings: 'Refs' might be clearer to remove and just state 'First author, year'

'Thresholds based on' - I am presuming this also relates to the parameters chosen as well as the thresholds applied so may need rewording

Underlying components: Not sure what 'underlying' means.

Brighton tools: Number of items in the tool is noted as 3, but there are also the nebs and vomiting categories, which would appear to qualify as a separate 'item' (so I would say total is 5). If there is a reason these are not included, it needs clarifying.

It would be really helpful to present the total number of component parameters at the bottom of the table so you can get a sense of what is most commonly used.

Table 3: suffers from similar issues of being very long and very wordy. I do feel you could lose some of the text in the column 'Notes on accuracy/reliability....'

In particular you have a column for 'PTTT used in practice', but repeatedly note in the description 'PTTT used in practice' which is unnecessary. I wonder if you could add a column for

Accuracy/reliability of scoring described in the same way, as that would also cut out a lot of text. Similarly 'no details on missing data' comes up a lot. Your wording in this column could be a lot tighter to make it more digestible. For example Page 46, Lines 10-17 could be condensed. The information is good, but you just feel overwhelmed with the length of it and it gets a bit repetitive to read.

Outcome measures: page 10 line 56 onwards

Largely this reads well but there are some sections where the style seems to change. So the Brighton PEWS and derivatives does not reference the 11 studies at the beginning like you do elsewhere, and the style of writing (use of authors names rather than describing the study characteristics) feels different to other sections. Its also quite long and more wordy compared to other sections, particular page 12 lines 12-23. The section on inter-rater reliability (page 13 lines 13-35) also makes lots of references to the authors which feels different to the style used elsewhere. Similarly page 16, lines 18-32.

The referencing of the studies at the beginning of each sub-section is really helpful but some of you references here don't match your tables. For example page 11, line 37 - reference 29 should be 74 according to your table. Page 42 line 25 should be reference 74, not 6. Its a lot of data, but it just needs checking.

Page 13 line 54 would benefit from the references being included and make it consistent with style elsewhere

Supplemental Table 4 and 5: Would be helpful to have consistency with the headings so 'refs' and 'Reference' would be 'First author, year'. If the ordering of columns could be as close to the ordering of columns in your main paper it would help with reader

orientation. Supp Table 5 is again quite wordy and would benefit from condensing if possible.

Page 14 - 15: there are some results which you describe as significant but where the 95% confidence interval straddles either side of 1, so would be considered not significant. Page 14, line 36; page 15 lines 11, 22, 44,46; As a result you need to reconsider your presentations of the findings in light of this. Similarly Table 4 presents many findings described as significant that are not: page 61, lines 31-35; page 62 lines 12-17; page 64 lines 5-20. Again, the table is quite repetitive to read as you put a narrative statement in the description of findings which I don't think is needed. I think it is sufficient to highlight the significant findings in bold as the reader can see the denominator, the outcome and the results making the narrative largely redundant. For some (page 57: line 5-10) you say there is non-significant trend. To my mind, this confuses the reader as, at the end of the day, the results remain not significant (and with a very wide CI in this case). Unless there is some other reason to treat these results differently to the others where the CI crosses 1, they remain non-significant. If you do treat them differently, you need to say why.

Discussion:

Overall a fair and balance discussion however this does need to be looked at in light of the finding reported in your results which do not reach significance. I have not crossed referenced each one to check whether it alters your message but it needs examining.

Page 17, lines 16 - 23: I am not sure what this very long and complex sentence is trying to say !!

Page 17 line 43 - both missing and excluded parameters may result in underscoring of the PTTT score (it will never overscore as far as I can see so you can be more specific).

page 19 line 11 - reference needed for the 3 cases.

Limitations: Helpful to explain why you did not conduct any meta-analysis.

Conclusion: Again, needs to be checked in light of the non-significant results, but I suspect your underlying message will remain the same. line 51 'unreliable availability' is a bit confusing, page 20 line 4-7 - this also takes a few reads to understand what you are saying.

Line 12: I'm not sure the word 'indeed' is the best one to start your final sentence - it made me feel like there was something else to follow this that I had missed.

Minor points - some spelling mistakes dotted around - page 14 line 10 interventions not nterventions; page 19 line 17: several Some abbreviations not described (or I have missed it, I apologise) - page 14, line 25 CI, line 35 OR, page 16 line 3 - IRR, TTT page 19, line 5

References need checking as some are in twice (Chapman, Monaghan) and some are incomplete - references 41, 65, ? 66. Could not find the PRISMA checklist (but I may have missed it as its a long document)

Overall a huge amount of work, largely well written and presented. My main concerns are

	the length and amount of detail in the tables - it feels too much to 'get your head around' and results in formatting whereby you cant see which PTTT you are reading about some results presented as significant which are not describing/justifying your methodology as PRISMA is a checklist for writing up, not a method in itself YOur method of quality assessment of the papers, given it appears to have been extensively modified from a 20 year old publication Overall I very much enjoyed reviewing it. Many thanks
--	---

REVIEWER	Jacqueline Birks University of Oxford, UK
REVIEW RETURNED	22-Oct-2018

GENERAL COMMENTS	The authors have carried out a major revision in response to detailed reviewer feedback. The additional tables and enlarged tables now report important information on all included studies. The inclusion of a quality score is a very important improvement . The reader can now assess the overall quality of the studies, and also for each PTTT. The higher quality studies are now easily identified. The strengths and limitations section is a good summary of the issues. The authors have brought to our attention the frequency with which the PTTT were derived using expert opinion and not statistical methods. . They note that complex statistical methods which can included longitudinal patient data have not been routinely used in deriving the risk scores, that further work on the methodology is needed. They conclude by warning on the automatic adoption of a PTTT and have backed this with excellent evidence. The review is an excellent summary of the evidence for PTTTs.
--

VERSION 2 – AUTHOR RESPONSE

Reviewer 3

Page 7, Line 6-8: I would question whether PRISMA is a methodology to be followed when conducting a SR. It is a minimum set of items for reporting a SR, but does not give guidance on conducting one. GRADE is acknowledged by many journals as the most robust method of conducting a SR. GRADE also allows recommendations to made regarding the strength of the literature as a whole rather than just the individual component studies. It would help the reader if you could justify why you chose this method rather than GRADE or other more recognised SR methodologies.

As above, we have amended the line in the methods section to clarify that PRIMSA guidelines were used to inform reporting, rather than the methodology for the review.

We acknowledge the benefits of the GRADE approach. We would note that is not so much a “method of conducting a SR” as a way for reviewers and guideline developers to assess the quality of the evidence and decide whether to recommend a particular intervention. While we did not use the GRADE approach, we have nonetheless evaluated and reported on the quality of evidence for each of the outcome measures and made recommendations for practice accordingly. The protocol for the study (including details of our approach to conducting the review) has been peer-reviewed and

published online (*). In a topic area characterised by methodological limitations and an absence of evidence, we don't believe that using the GRADE methodology would have changed our conclusions or recommendations in this instance.

P7, L 53: acuity at PICU admission and PICU outcomes stated here but not on PICOS (Supplementary table 2)

We have added these outcomes to the PICOS table (Supplementary Table 2).

P8 L21: Mainly reported 95% CI not p values. Also reported OR and other analyses which need including here. Your description of data analysis feels a little sparse.

Have updated this section on data analysis for clarity and to add more detail.

Quality appraisal: This is probably my main criticism of the conduct of the study. Downs and Black is now 20 years old and it is not clear why you chose this above other quality measures. It is also not clear how this was modified (the original paper reports 27 items but your modified tool contains 12. This links with my feedback above regarding the methodology chosen. Assessing the quality of individual papers, whilst helpful, does not make it clear to the reader the strength of the evidence 'as a whole' and limits the ability to truly synthesise the literature.

Although the checklist was originally published in 1998, it has not been discredited and continues to be used as a means of quality appraisal by researchers conducting systematic reviews of health interventions (e.g., (Kominiarek, Lewkowitz, Carter, Fowler, & Simon, 2019; Siaw & Lee, 2018)). . Our decision to use this appraisal checklist was pre-specified in our Prospero protocol and in our peer-reviewed protocol for the wider study (Thomas-Jones et al., 2018). The amended version of the checklist is included in Supplementary Table 3 to allow readers to see the exact criteria used for evaluating studies.

We believe that evaluation of individual papers served an important function here, by highlighting the overall scarcity of high-quality studies in this area and spotlighting the handful of robustly designed studies. As above, we believe we used these quality appraisals in combination with the effect sizes reported for each outcome to give a clear overview of the strength of evidence for PTTTs for reducing various indices of in-hospital deterioration.

Table 1: The section on study design does not present items in rank order which is inconsistent with the rest of the table.

This has been amended accordingly.

Table 2: This is probably my other area of concern. Your tables are really long and quite labour intensive on the part of the reader. If it is at all possible to reduce down the wording, particularly in the 3rd column, it would make it more digestible. The information is helpful, but you just get a bit overloaded.

We have condensed the information in this column as much as possible, which will hopefully improve readability. We have had to be conscious that much of this extra level of detail was previously requested by other reviewers!

In terms of headings: 'Refs' might be clearer to remove and just state 'First author, year'

Agreed. We have actually removed the author names and used reference numbers, again to help reduce the height of the rows and improve readability.

'Thresholds based on' - I am presuming this also relates to the parameters chosen as well as the thresholds applied so may need rewording

Underlying components: Not sure what 'underlying' means.

The wording of each heading has been updated accordingly ("Choice of thresholds / parameters" & "PTTT parameters" respectively)

Brighton tools: Number of items in the tool is noted as 3, but there are also the nebs and vomiting categories, which would appear to qualify as a separate 'item' (so I would say total is 5). If there is a reason these are not included, it needs clarifying.

Agreed – updated to 5.

It would be really helpful to present the total number of component parameters at the bottom of the table so you can get a sense of what is most commonly used.

This is challenging to add to the table itself, but the most commonly used parameters are reported in the main text:

“Common parameters included heart rate (present in 26 out of 27 PTTT), respiratory rate (24), respiratory effort (24) and level of consciousness or behavioural state (24)”

Table 3: suffers from similar issues of being very long and very wordy. I do feel you could lose some of the text in the column 'Notes on accuracy/reliability....

In particular you have a column for 'PTTT used in practice', but repeatedly note in the description 'PTTT used in practice' which is unnecessary. I wonder if you could add a column for Accuracy/reliability of scoring described in the same way, as that would also cut out a lot of text. Similarly 'no details on missing data' comes up a lot. Your wording in this column could be a lot tighter to make it more digestible. For example Page 46, Lines 10-17 could be condensed. The information is good, but you just feel overwhelmed with the length of it and it gets a bit repetitive to read.

Again, we have condensed the information in this 'notes' column as far as possible, while trying to maintain the extra information requested by other reviewers.

Outcome measures: page 10 line 56 onwards

Largely this reads well but there are some sections where the style seems to change. So the Brighton PEWS and derivatives does not reference the 11 studies at the beginning like you do elsewhere, and the style of writing (use of authors names rather than describing the study characteristics) feels different to other sections. Its also quite long and more wordy compared to other sections, particular page 12 lines 12-23. The section on inter-rater reliability (page 13 lines 13-35) also makes lots of references to the authors which feels different to the style used elsewhere. Similarly page 16, lines 18-32.

Both sections have been re-worded and simplified, in line with the style of the other sections mentioned.

The referencing of the studies at the beginning of each sub-section is really helpful but some of you references here don't match your tables. For example page 11, line 37 - reference 29 should be 74 according to your table. Page 42 line 25 should be reference 74, not 6. Its a lot of data, but it just needs checking.

We have checked these references and updated them (and the references section) accordingly – this was caused by duplication of a couple of references in the reference manager.

Page 13 line 54 would benefit from the references being included and make it consistent with style elsewhere

The seventeen references have been added.

Supplemental Table 4 and 5: Would be helpful to have consistency with the headings so "refs" and 'Reference' would be 'First author, year'. If the ordering of columns could be as close to the ordering of columns in your main paper it would help with reader orientation.

Headings have been changed to "First author, year" accordingly. Supplemental Table 4 column headers mirror those in Table 3 of the main paper, while Supplemental Table 5 column headers mirror those in Table 4.

Supp Table 5 is again quite wordy and would benefit from condensing if possible.

Have trimmed some of the detail in the penultimate column.

Page 14 - 15: there are some results which you describe as significant but where the 95% confidence interval straddles either side of 1, so would be considered not significant. Page 14, line 36; page 15 lines 11, 22, 44,46; As a result you need to reconsider your presentations of the findings in light of this. Similarly Table 4 presents many findings described as significant that are not: page 61, lines 31-35; page 62 lines 12-17; page 64 lines 5-20

We have made minor amendments to Table 4 accordingly. For the Parshuram 2011 and Brill 2007 papers, we have recalculated p-values based on the raw numbers presented in the paper and corresponding RR values, as those reported in the original paper appear to be incorrect. We have also re-calculated and corrected the effect size and p-value for the Hunt 2008, as again, the values reported in the original papers appear incorrect. All other effect size and p-values have been re-checked but are unchanged.

The corresponding text in the results section of the paper has all been update to reflect the amended values for the three affected papers.

Again, the table is quite repetitive to read as you put a narrative statement in the description of findings which I don't think is needed. I think it is sufficient to highlight the significant findings in bold as the reader can see the denominator, the outcome and the results making the narrative largely redundant. For some (page 57: line 5-10) you say there is non-significant trend. To my mind, this confuses the reader as, at the end of the day, the results remain not significant (and with a very wide CI in this case). Unless there is some other reason to treat these results differently to the others where the CI crosses 1, they remain non-significant. If you do treat them differently, you need to say why.

We have removed the column referred to here, giving a narrative description of results. We agree that this is largely redundant, and have made bold all of the findings which reach statistical significance.

Discussion:

Overall a fair and balance discussion however this does need to be looked at in light of the finding reported in your results which do not reach significance. I have not crossed referenced each one to check whether it alters your message but it needs examining.

We removed one reference (Parshuram, 2011) from the sentence discussing effectiveness studies which refers to "several other studies reported significant increases in transfers or calls for review", but otherwise the Discussion section remains unchanged as a result of the abovementioned changes.

Page 17, lines 16 - 23: I am not sure what this very long and complex sentence is trying to say !!

We have removed this sentence and tweaked the wording of the sentences around it – hopefully the message is clearer.

Page 17 line 43 - both missing and excluded parameters may result in underscoring of the PTTT score (it will never overscore as far as I can see so you can be more specific).

Agreed, and have added to the sentence accordingly.

page 19 line 11 - reference needed for the 3 cases.

References added.

Limitations: Helpful to explain why you did not conduct any meta-analysis.

Have added this point to the end of the limitations section of the Discussion.

Conclusion: Again, needs to be checked in light of the non-significant results, but I suspect your underlying message will remain the same

As indicated, the amendments do not change the underlying message here.

line 51 'unreliable availability' is a bit confusing

Have changed “unreliable availability of many underlying observations” to “many PTTT parameters not being reliably recorded in practice”.

page 20 line 4-7 - this also takes a few reads to understand what you are saying

Have amended the wording here to improve readability.

Line 12: I'm not sure the word 'indeed' is the best one to start your final sentence - it made me feel like there was something else to follow this that I had missed.

Have removed 'indeed' and reworded this sentence.

Minor points - some spelling mistakes dotted around - page 14 line 10 interventions not nterventions;
page 19 line 17: several

Both typos corrected.

Some abbreviations not described (or I have missed it, I apologise) - page 14, line 25 CI, line 35 OR,
page 16 line 3 - IRR, TTT page 19, line 5

Have gone back to check that these abbreviations are spelled out in full where they first appear in the manuscript.

References need checking as some are in twice (Chapman, Monaghan) and some are incomplete - references 41, 65, ? 66.

We have corrected the duplications and tidied up the incomplete references – apologies, these were reference manager errors.

Could not find the PRISMA checklist (but I may have missed it as its a long document)

As above, this was mistakenly omitted from the re-submission, but it is now included along with the other supplemental documents in the new submission.

References

Kominiarek, M. A., Lewkowitz, A. K., Carter, E., Fowler, S. A., & Simon, M. (2019). Gestational weight gain and group prenatal care: a systematic review and meta-analysis. *BMC Pregnancy and Childbirth*, 19(1), 18. <http://doi.org/10.1186/s12884-018-2148-8>

Siaw, M. Y. L., & Lee, J. Y.-C. (2018). Multidisciplinary collaborative care in the management of patients with uncontrolled diabetes: A systematic review and meta-analysis. *International Journal of Clinical Practice*, e13288. <http://doi.org/10.1111/ijcp.13288>

Thomas-Jones, E., Lloyd, A., Roland, D., Sefton, G., Tume, L., Hood, K., ... Powell, C. (2018). A prospective, mixed-methods, before and after study to identify the evidence base for the core components of an effective Paediatric Early Warning System and the development of an implementation package containing those core recommendations for use in the UK: Paediatric early warning system - utilisation and mortality avoidance- the PUMA study protocol. *BMC Pediatrics*, 18(1), 244. <http://doi.org/10.1186/s12887-018-1210-z>

VERSION 3 – REVIEW

REVIEWER	Dr Sue Chapman International and Private Patients Division, Great Ormond Street Hospital for Children London, UK
REVIEW RETURNED	11-Feb-2019

GENERAL COMMENTS	Thank you for inviting me to review this manuscript. It has improved considerably and I thank the authors for their revisions. The manuscript is well written, clear and makes a valuable contribution to the field. The Tables and Figures are clear. Discussion is thoughtful. Conclusion is balanced. I have only relatively minor comments to make. Table 1: page 11 line 4 – 19,4 or 19.4? Page 18 line 18 – minor typo of full stops. Some references need attention as they look incomplete – in particular 41, 65, 8- and 50 is a duplicate of 29. Table 2: not quite clear how these are ordered or grouped. If it is based on the tool they were derived from a sub-heading may help. Also be helpful to have abbreviations presented in alphabetical order to assist with cross-referencing. Table 3: again, alphabetical order of abbreviations would assist and inclusion of ‘max’ (presumably maximum from the column ‘which score used’) and NA and N/A – not sure if these represent different things but both are used. Table 4: In the column PTTT not sure if you need the abbreviation for each PTTT as you then write in full subsequently. There is no explanation of what items in bold in the p value column mean (presumably that they are significant). Some items in the numbers of centres need reformatting as you could read the item on page 45 lines 28-34 as 2 and 1 centres rather than 21. Same for line 35-38 – is it 38 or 3 and 8? Page 54 lines 8-15 need to explain ‘con’ and ‘int’ in abbreviations. IRR is also not explained. I like the formatting of the table overall - clearly presented. Page 58 – The relevance of PUMA is not clear.
--

	Search strategies: It would help to have the name of the database presented in full as the heading rather than BNI, Central etc. Page 74 line 30 – receiver operator not operating. Again alphabetic order of abbreviations would help. Also help to have the abbreviations for the PTTT included as Table should stand alone and not require cross-referencing to another Table in a different section. Many thanks for inviting me to review this manuscript. I enjoyed reading it.
--	--

VERSION 3 – AUTHOR RESPONSE

Reviewer 3

Table 1: page 11 line 4 – 19,4 or 19.4?

Page 18 line 18 – minor typo of full stops.

Both were typos, now corrected.

Some references need attention as they look incomplete – in particular 41, 65, 8- and 50 is a duplicate of 29.

References in question have been checked and updated accordingly – duplicate reference removed and codes updated.

Table 2: not quite clear how these are ordered or grouped. If it is based on the tool they were derived from a sub-heading may help. Also be helpful to have abbreviations presented in alphabetical order to assist with cross-referencing.

Abbreviations are now in alphabetical order (for this, and all other tables).

The table is ordered by the PTTT that they were derived from, and we've now added sub-headings to clarify.

Table 3: again, alphabetical order of abbreviations would assist and inclusion of 'max' (presumably maximum from the column 'which score used/') and NA and N/A – not sure if these represent different things but both are used.

Abbreviations updated accordingly, and instances of NA changed to N/A for not applicable.

Table 4: In the column PTTT not sure if you need the abbreviation for each PTTT as you then write in full subsequently. There is no explanation of what items in bold in the p value column mean (presumably that they are significant). Some items in the numbers of centres need reformatting as you could read the item on page 45 lines 28-34 as 2 and 1 centres rather than 21. Same for line 35-38 – is it 38 or 3 and 8? Page 54 lines 8-15 need to explain 'con' and 'int' in abbreviations. IRR is also not explained. I like the formatting of the table overall - clearly presented.

PTTT abbreviations have been removed.

We have added an explanation in the table footer that p-values in bold represent statistical significance (<0.05).

Numbers in the 'number of centers' column should now display correctly, having adjusted column width.

Con, Int and IRR abbreviations added to the footer.

Page 58 – The relevance of PUMA is not clear.

Search strategies: It would help to have the name of the database presented in full as the heading rather than BNI, Central etc.

The word PUMA has been removed from the heading and the database names have now been presented in full.

Page 74 line 30 – receiver operator not operating. Again alphabetic order of abbreviations would help. Also help to have the abbreviations for the PTTT included as Table should stand alone and not require cross-referencing to another Table in a different section.

Typo corrected, and abbreviations for this and all other tables are now in alphabetical order. PTTT names have been updated accordingly.